# A two-dimensional Fe-doped SnS$_2$ magnetic semiconductor

Bo Li[1,2], Tao Xing[3], Mianzeng Zhong[1], Le Huang[4], Na Lei[3], Jun Zhang [1], Jingbo Li[1] & Zhongming Wei [1]

Magnetic two-dimensional materials have attracted considerable attention for their significant potential application in spintronics. In this study, we present a high-quality Fe-doped SnS$_2$ monolayer exfoliated using a micromechanical cleavage method. Fe atoms were doped at the Sn atom sites, and the Fe contents are ~2.1%, 1.5%, and 1.1%. The field-effect transistors based on the Fe$_{0.021}$Sn$_{0.979}$S$_2$ monolayer show n-type behavior and exhibit high optoelectronic performance. Magnetic measurements show that pure SnS$_2$ is diamagnetic, whereas Fe$_{0.021}$Sn$_{0.979}$S$_2$ exhibits ferromagnetic behavior with a perpendicular anisotropy at 2 K and a Curie temperature of ~31 K. Density functional theory calculations show that long-range ferromagnetic ordering in the Fe-doped SnS$_2$ monolayer is energetically stable, and the estimated Curie temperature agrees well with the results of our experiment. The results suggest that Fe-doped SnS$_2$ has significant potential in future nanoelectronic, magnetic, and optoelectronic applications.

[1] State Key Laboratory of Superlattices and Microstructures, Institute of Semiconductors, Chinese Academy of Sciences & College of Materials Science and Opto-Electronic Technology, University of Chinese Academy of Sciences, Beijing 100083, China. [2] Department of Applied Physics, School of Physics and Electronics, Hunan University, Changsha 410082, China. [3] Fert Beijing Institute, School of Electronic and Information Engineering, BDBC, Beihang University, Beijing 100191, China. [4] School of Materials and Energy, Guangdong University of Technology, Guangzhou, Guangdong 510006, China. Correspondence and requests for materials should be addressed to Z.W. (email: zmwei@semi.ac.cn)

Two-dimensional (2D) layered transition metal dichalco-genides (TMDs), e.g., $MoS_2$, $WS_2$, and $SnS_2$, are promising functional materials due to their peculiar structural and electronic properties[1–10]. Significant efforts, such as doping, strain, and chemical functionalization, have been used to obtain distinctive optical and electrical properties by tuning the band alignments of 2D materials[11–14]. Doping, which is the intentional introduction of impurities into a material, plays a significant role in functionalizing 2D materials by changing the intrinsic properties of pristine atomic layers[15–17]. For example, wolfram and selenium chemical doping of $MoS_2$ is an effective way to engineer the optical bandgap[18–23], and Nb-, Co-, and Mn-doped $MoS_2$ few layers exhibit diverse transport properties[11, 12, 24]. Magnetic atom (e.g., Mn, Fe, Co, and Ni)-doped 2D TMDs are promising as 2D-diluted magnetic semiconductors (DMS)[25–29], and many have been predicted to exhibit ferromagnetic behavior at room temperature[25, 27]. DMS, such as Mn-doped InAs and GaAs, have distinctive physical properties and provide the possibility of electronic control of magnetism[30–33]. To date, Co- and Mn-doped $MoS_2$ nanosheets have been synthesized via the chemical vapor deposition method, and understanding their magnetic properties requires more research[11, 24]. Recently, Zhang et al. and Xu et al. reported the magnetic properties of $Cr_2Ge_2Te_6$ and $CrI_3$ monolayers via high-resolution and high-sensitivity magneto-optic microscopy[34, 35]. However, further investigation into the magnetism and functional properties of high-quality magnetic atom-doped 2D TMDs is warranted.

In this work, we synthesized different Fe-doped $SnS_2$ ($Fe_{0.021}Sn_{0.979}S_2$, $Fe_{0.015}Sn_{0.985}S_2$, and $Fe_{0.011}Sn_{0.989}S_2$) bulk crystals via a direct vapor-phase method, and we obtained $Fe_{0.021}Sn_{0.979}S_2$ monolayer flakes via mechanical exfoliation. Monolayer $Fe_{0.021}Sn_{0.979}S_2$ exhibits high-quality optoelectronic properties. Magnetic measurements show that $Fe_{0.021}Sn_{0.979}S_2$ exhibits ferromagnetic behavior with a perpendicular anisotropy at 2 K and a Curie temperature of ~31 K. The experimental results agree well with the theoretical calculations.

## Results

**Characterization of Fe-doped $SnS_2$.** Figure 1a shows an optical image of a mechanically exfoliated $Fe_{0.021}Sn_{0.979}S_2$ flake on a Si/$SiO_2$ substrate. Atomic force microscopy (AFM) and magnetic force microscopy (MFM) were used to characterize the height and magnetism of the samples, respectively. MFM is a valuable tool that can potentially be used to detect magnetic interactions between a magnetized AFM tip and samples[36]. Recently, MFM has been employed to characterize the magnetic response of single- or few-layer 2D nanosheets, such as graphene and $MoS_2$[37, 38]. However, it has been reported that MFM signals have non-magnetic contributions due to capacitive and electrostatic interactions between the nanosheets and the conductive cantilever tip[38, 39]. In this study, AFM and MFM images of $Fe_{0.021}Sn_{0.979}S_2$ and pure $SnS_2$ were obtained under the same test condition. The AFM images show that the obtained $Fe_{0.021}Sn_{0.979}S_2$ and pure $SnS_2$ are monolayers (Fig. 1b, Supplementary Fig. 1a). The MFM images show that the $Fe_{0.021}Sn_{0.979}S_2$ monolayer has a larger negative phase shift (523 m°) than that of the pure $SnS_2$ monolayer (51 m°) (Supplementary Fig. 1b, c) by approximately ten times, which should largely contribute to the difference in the magnetic and electrical properties between the $Fe_{0.021}Sn_{0.979}S_2$ and $SnS_2$ monolayers. Raman spectroscopy has been widely used in 2D TMD alloys, and it changes with the composition of the alloy[18, 40]. The Raman peaks of the $Fe_{0.021}Sn_{0.979}S_2$ and pure $SnS_2$ monolayers are located at 314 cm$^{-1}$, corresponding to the $A_{1g}$ mode of $SnS_2$[3, 10]. The $A_{1g}$ mode of the $Fe_{0.021}Sn_{0.979}S_2$ monolayer is broader than that of the pure $SnS_2$ monolayer (Fig. 1c). This broadening behavior, which is due to the doped atoms, has also been observed in other 2D alloys[20].

The crystallinity of the $Fe_{0.021}Sn_{0.979}S_2$ was further characterized using high-angle annular dark-field scanning transmission electron microscopy (HAADF-STEM) and transmission electron microscopy (TEM). Figure 1e shows a low-resolution HAADF-STEM image of a few layers of $Fe_{0.021}Sn_{0.979}S_2$. Energy-dispersive X-ray spectroscopy (EDS) of the nanosheet (Fig. 1f) shows that

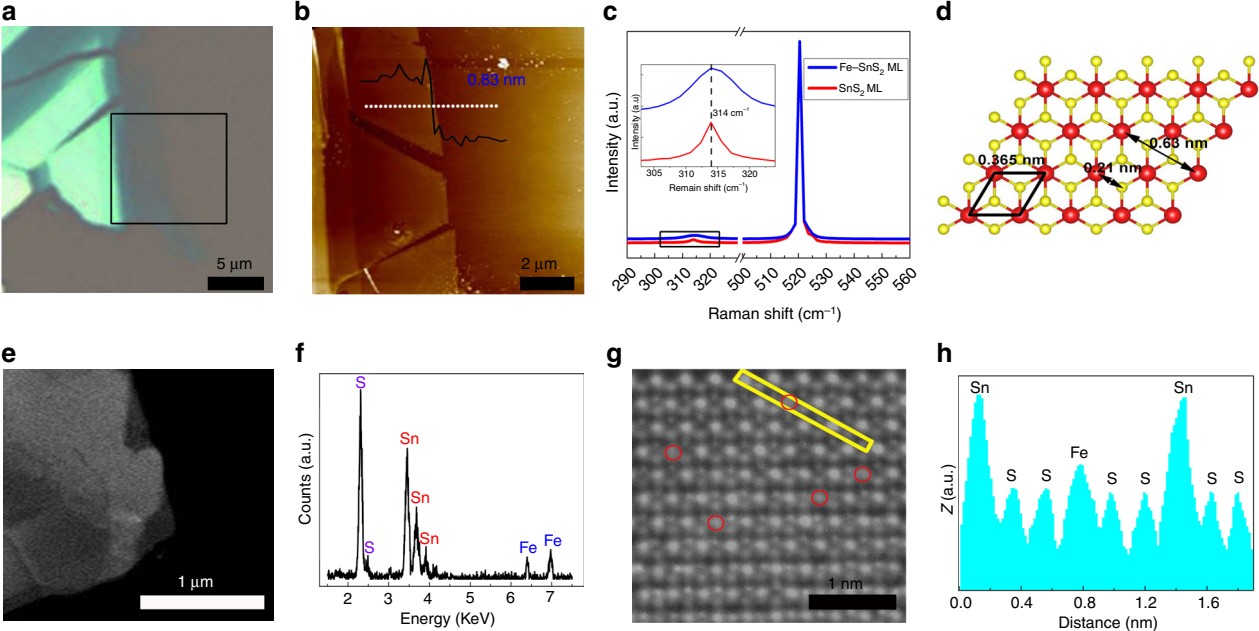

**Fig. 1** Characterization of the $Fe_{0.021}Sn_{0.979}S_2$ flakes. **a** Optical image of the $Fe_{0.021}Sn_{0.979}S_2$ flake. **b** AFM and **c** Raman spectra of the $Fe_{0.021}Sn_{0.979}S_2$ and $SnS_2$ monolayers. The height of the flake in **b** was obtained along the white dotted line. The inset in **c** shows the expanded view of the Raman spectra around the $A_{1g}$ mode of the $Fe_{0.021}Sn_{0.979}S_2$ and $SnS_2$ monolayers. **d** $SnS_2$ atomic structure. **e** Low-resolution HAADF-STEM image of the $Fe_{0.021}Sn_{0.979}S_2$ flake. **f** EDS of the $Fe_{0.021}Sn_{0.979}S_2$ flake. **g** High-resolution STEM image of the $Fe_{0.021}Sn_{0.979}S_2$ flake; the red circles are Fe atoms. **h** Z-contrast mapping in the areas marked with yellow rectangles in **g**

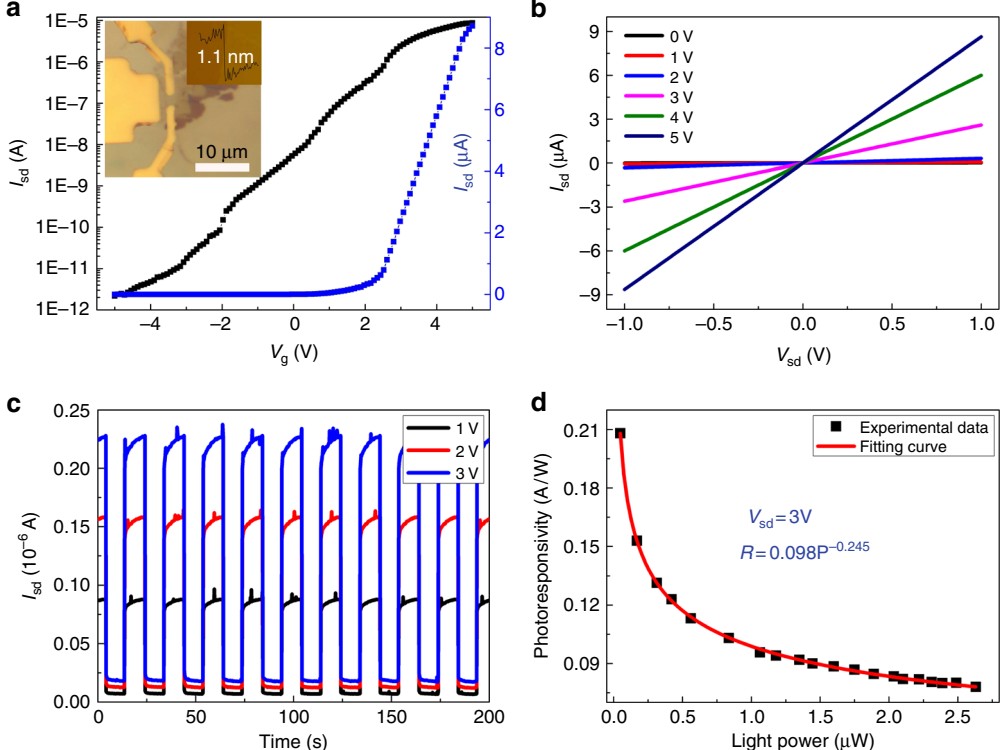

**Fig. 2** Electrical characteristics and photoresponse of the $Fe_{0.021}Sn_{0.979}S_2$ monolayer. **a** Transfer and **b** output characteristics of $Fe_{0.021}Sn_{0.979}S_2$. The inset shows an optical image of one typical device and the AFM image of the corresponding sample used for fabricating the device. **c** Time-dependent $I_{sd}$ of the transistor based on $Fe_{0.021}Sn_{0.979}S_2$ during the light (638 nm, 2.63 μW) switching on/off under a positive source–drain voltage, $V_{sd}$, from 1 to 3 V. **d** Photoresponsivity ($R$) as a function of light power ($P$) with a $V_{sd}$ of 3 V

the nanosheet contains S, Sn, and Fe elements. Using a high-resolution STEM image and the Z-contrast intensity distribution, the Sn ($Z = 50$), Fe ($Z = 26$), and S ($Z = 16$) atoms were directly distinguished, as shown in Fig. 1g, h. Fe atoms were doped at the Sn atom sites. The Sn–S distance is 0.22 nm (Fig. 1h), which agrees well with the theoretical value of 0.21 nm (Fig. 1d). The low-resolution TEM image (Supplementary Fig. 2a) shows a part of the few-layer $Fe_{0.021}Sn_{0.979}S_2$ flake on the holey carbon TEM grid. The selected area electron diffraction pattern and the corresponding high-resolution TEM image reveal that this flake has a high-quality hexagonal symmetry structure, and the lattice spacing of (100) plane is 0.311 nm (Supplementary Fig. 2b, c). The elemental mapping images from EDS show that Sn, S, and Fe elements are uniformly distributed throughout the entire flake (Supplementary Fig. 2d–f).

X-ray photoelectron spectroscopy (XPS) is a powerful tool for understanding the chemical states and composition of elements that exist within a material. The binding energy values obtained in the XPS analysis of $Fe_{0.021}Sn_{0.979}S_2$ were corrected by referencing the C 1s peak to 284.7 eV (Supplementary Fig. 3a). The binding energies of the Sn $3d_{5/2}$ and S $2p_{3/2}$ electron peaks are 486.7 eV and 161.6 eV (Supplementary Fig. 3b, c), respectively, which is consistent with the previously reported values for $SnS_2$[41]. The binding energy of the Fe $2p_{3/2}$ electron peak is 712.2 eV (Supplementary Fig. 3d), which is close to the Sn $3p$ electron peak (716 eV). The binding energy of the Fe $2p_{3/2}$ electron in this instance is obviously larger than the reported values of other iron compounds (the binding energies of Fe $2p_{3/2}$ in $Fe_2O_3$, $FeCl_3$, and $FeCl_2$ are 710 eV, 711.5 eV, and 710.6 eV, respectively)[42]. In general, the binding energy of the same electron from one atom grows with increasing oxidation state, which is related to the bonding hybridization with its nearest-neighbor atoms. In this experiment, Fe atoms were doped by substituting Sn sites

(Fig. 1g), and each Fe atom is surrounded by six S atoms to form an octahedral coordination. The oxidation state of the Fe atom should be +4, leading to a high binding energy for Fe $2p_{3/2}$ in $Fe_{0.021}Sn_{0.979}S_2$. Furthermore, quantitative analysis of the XPS spectra reveals that the content of Fe is ~2.1% in $Fe_{0.021}Sn_{0.979}S_2$.

**Electronic and optoelectronic properties of Fe-doped $SnS_2$.** To study the electronic transport property and photoresponse of the $Fe_{0.021}Sn_{0.979}S_2$ monolayer on the $Al_2O_3$/Si substrate, we fabricated field-effect transistors (FETs) from exfoliated $Fe_{0.021}Sn_{0.979}S_2$ (Fig. 2a). The device characteristics of the FETs were measured at room temperature. Figure 2a, b shows the typical transfer and output characteristics of $Fe_{0.021}Sn_{0.979}S_2$, respectively, achieving an excellent n-type behavior. As the gate voltage ($V_g$) varied from −5 to 5 V, the source–drain current ($I_{sd}$) changed from $1.1 \times 10^{-12}$ A to $8 \times 10^{-6}$ A, corresponding to a high on/off current ratio of $7.3 \times 10^6$. The field-effect mobility can be obtained using the formula

$$\mu = \frac{\partial I_{sd}}{\partial V_g}\left(\frac{L}{WC(Al_2O_3)V_{sd}}\right), \qquad (1)$$

where $L$ and $W$ are the length and width of the device, and $C(Al_2O_3)$ is the $Al_2O_3$ gate capacitance, which can be given by equation $C(Al_2O_3)=\varepsilon_0\varepsilon_r/d$. Thus, $\varepsilon_0$ ($8.85 \times 10^{-12}$ Fm$^{-1}$) is the vacuum dielectric constant, and $\varepsilon_r$ (6.4) and $d$ (30 nm) are the dielectric constant and thickness of $Al_2O_3$, respectively. Based on the transport curve (Fig. 2a), the calculated electron mobility is 8.15 cm$^2$ V$^{-1}$ s$^{-1}$. $Fe_{0.021}Sn_{0.979}S_2$ and $Fe_{0.021}Sn_{0.979}S_2$ were also synthesized by modulating the growth conditions (Supplementary Fig. 4). The electronic transport properties of $Fe_{0.021}Sn_{0.979}S_2$ (Supplementary Fig. 5) were investigated, and the calculated mobilities are larger than the values for pure $SnS_2$

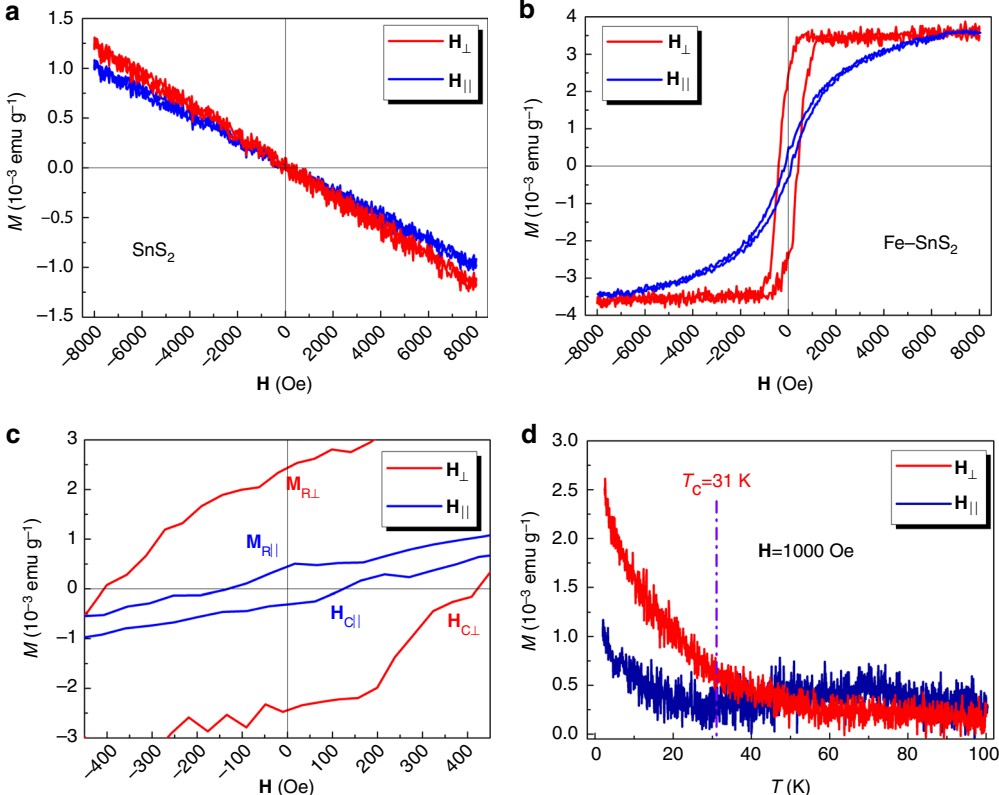

**Fig. 3** Magnetization data for $SnS_2$ and $Fe_{0.021}Sn_{0.979}S_2$. **a, b** Magnetic hysteresis loops for $SnS_2$ and $Fe_{0.021}Sn_{0.979}S_2$ at 2 K using VSM, respectively. **c** Expanded view of the loop of $Fe_{0.021}Sn_{0.979}S_2$ in **b**. **d** Magnetization as a function of temperature for $Fe_{0.021}Sn_{0.979}S_2$ from 2 K to 100 K. The applied magnetic field was 1000 Oe

(Supplementary Table 1). The mobility increases with the Fe content in the samples. The output curves show that the $I_{sd}$ is linear at low $V_{sd}$, and the sample had a good contact with the electrode (Fig. 2b). The mobility is related to the mean-free time ($\tau$) and effective mass ($m^*$) of the electron as follows:

$$\mu = \frac{q\tau}{m^*}. \qquad (2)$$

In 2D-doped TMDs, the mean-free time of the electron is decreased for enhanced carrier scattering, and the mobility decreases. Thus, searching for a decreased electron effective mass for 2D-doped TMDs is a feasible route to improve the mobility. The density functional theory (DFT)-calculated $m^*$ values at the conduction band bottom of monolayer $SnS_2$ and $Fe–SnS_2$ are listed in Supplementary Table 2. The $m^*$ of $Fe–SnS_2$ is smaller than that of pure $SnS_2$, which partly contributes to the higher field-effect mobility (Supplementary Fig. 6). The stability of 2D atomic layer materials is critical for their future application. The electrical property of a typical $Fe_{0.021}Sn_{0.979}S_2$ monolayer FET on a $SiO_2/Si$ substrate stored in air was measured for 1 month. After 1 month, the mobility changed from 6.1 cm$^2$ V$^{-1}$ s$^{-1}$ to 4.7 cm$^2$ V$^{-1}$ s$^{-1}$, and the on/off ratio changed from $1.2 \times 10^6$ to $7 \times 10^5$ (Supplementary Fig. 7). The results show that $Fe_{0.021}Sn_{0.979}S_2$ is very stable and has a significant potential application in optoelectronics.

Few-layer $SnS_2$ has been demonstrated as ultrasensitive photodetectors based on previous reports[3]. In this study, the photoresponsive properties of the $Fe_{0.021}Sn_{0.979}S_2$ monolayer were examined using a 638-nm laser at room temperature. Figure 2c shows a photo on/off ratio of ~10, and the $I_{sd}$ can quickly and repetitively change between on and off states. The

photoresponsivity, $R$, was obtained by using the formula $R = I_{ph}/P$. $I_{ph}$ is the photocurrent defined as $I_{ph} = I_{light}–I_{dark}$, and $P$ is the light power. The photoresponsivity ($R$) shows a strong dependence on light power ($P$), and the experimental data are fitted by the equation $R = aP^{\alpha-1}$. In our experiment, the fitted parameters were $a = 0.098$ and $\alpha = 0.755$ (Fig. 2d). The maximum $R$ is 206 mA W$^{-1}$ ($P = 57$ nW), which is larger than the reported average values of few-layer $SnS_2$ (Supplementary Table 1). The photocurrent response time of the $Fe_{0.021}Sn_{0.979}S_2$ is ~9 ms, and the response time of pure $SnS_2$ is ~6 ms (Supplementary Fig. 8). Under illumination, the impurity levels will lightly promote electron–hole recombination and increase the response time. A detailed discussion of the optoelectronic properties of $Fe–SnS_2$ is provided in Supporting Information (Supplementary Fig. 9, Supplementary Note 1).

**Magnetic properties of the Fe-doped $SnS_2$.** The magnetic behaviors of the $SnS_2$ and $Fe_{0.021}Sn_{0.979}S_2$ single-crystal sheets were investigated using a vibrating sample magnetometer (VSM) from a physical properties measurement system (PPMS). The measurements were performed in two types of applied magnetic fields (**H**): perpendicular to the sheet, e.g., parallel to the [001] direction ($\mathbf{H_\perp}$), and parallel to the sheet, e.g., perpendicular to the [001] direction ($\mathbf{H_\parallel}$). Magnetic hysteresis loops at 2 K for $SnS_2$ and $Fe_{0.021}Sn_{0.979}S_2$ are shown in Fig. 3a, b. The pure $SnS_2$ is diamagnetic at 2 K both in the $\mathbf{H_\perp}$ and $\mathbf{H_\parallel}$ directions because of the saturated electronic structure. The M–H curve of the $Fe_{0.021}Sn_{0.979}S_2$ sheet shows remarkable anisotropy in the $\mathbf{H_\parallel}$ and $\mathbf{H_\perp}$ directions at 2 K, respectively. The saturation magnetization ($\mathbf{M_S}$), coercivity ($\mathbf{H_C}$), and remnant magnetization ($\mathbf{M_R}$) values in the $\mathbf{H_\perp}$ and $\mathbf{H_\parallel}$ direction for $Fe_{0.021}Sn_{0.979}S_2$ are listed in Supplementary Table 3. The coercivity and remnant magnetization of

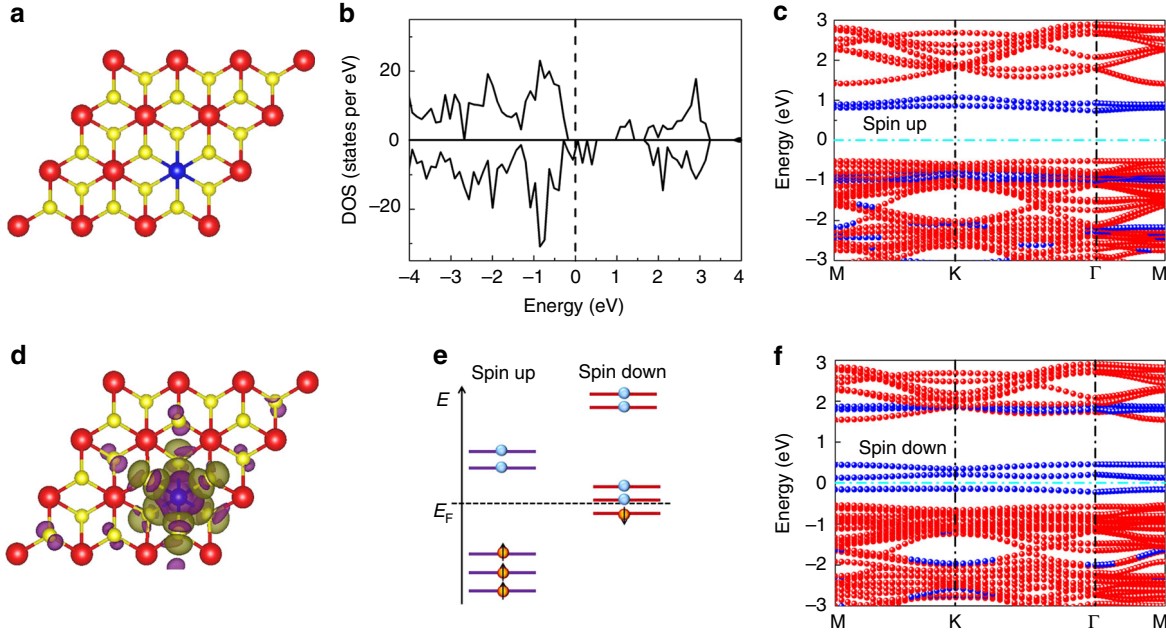

**Fig. 4** Theoretical calculations of the Fe–SnS$_2$ monolayer. **a** Atomic structure of the Fe–SnS$_2$ monolayer. Red, yellow, and blue balls represent Sn, S, and Fe atoms, respectively. **b** Total density of states of the Fe–SnS$_2$ monolayer. **c**, **f** Projected band structures of the Fe–SnS$_2$ monolayer for up-spin and down-spin channels, respectively. Red and blue circles denote the contribution of Sn and Fe atoms in the total band structure, respectively. The Fermi level was set to zero. **d** The distribution of the spin density in real space for the Fe–SnS$_2$ monolayer is shown. The isosurface value was taken at 0.001 eÅ$^{-3}$. **e** Schematic of the Fe 3$d$ electron arrangement with spin in Fe–SnS$_2$

$H_\perp$ are approximately three and five times those of $H_\parallel$, confirming that the easy axis is the [001]. The magnetism versus temperature curves show that the Curie temperature ($T_C$) is ~31 K (Fig. 3d). Magnetic hysteresis loops of the Fe$_{0.021}$Sn$_{0.979}$S$_2$ bulk crystal were acquired under different temperatures (Supplementary Fig. 10), and the result shows that the Curie temperature is between 35 K and 30 K, which is consistent with the magnetism versus temperature curves (Fig. 3d). Fe cluster can be formed in the crystal when the content of Fe source increases during growth (Supplementary Fig. 11 and Supplementary Note 2). The resistivity of the Fe$_{0.021}$Sn$_{0.979}$S$_2$ thin film as a function of different thicknesses was measured from 200 to 8 K (Supplementary Fig. 12). The temperature derivative of the measured resistivity has a transition at ~32 K for the monolayer and 41 K for 13- and 35-nm-thick samples, which are similar to the results observed at the Curie temperature of 31 K. This resistive transition could come from the magnetic transition. Fe$_{0.021}$Sn$_{0.979}$S$_2$ and Fe$_{0.015}$Sn$_{0.985}$S$_2$ are ferromagnetic at 2 K, whereas Fe$_{0.011}$Sn$_{0.989}$S$_2$ is paramagnetic at 2 K (Supplementary Fig. 13). The results demonstrate that the magnetic property of Fe–SnS$_2$ varies with the Fe concentration. We discuss the detailed origin of the magnetism and the magnetic anisotropy of the Fe$_{0.021}$Sn$_{0.979}$S$_2$ sheet below.

## Discussion

Wu et al. investigated the magnetism of bulk Fe-doped SnS$_2$ in detail and observed that the ferrimagnetism originates from the exchange interaction between the doped Fe atoms at the intralayer sites[27]. In this study, first-principles calculations were used to further investigate the electronic and magnetic properties of the Fe–SnS$_2$ monolayer (Fig. 4a). According to the generalized gradient approximation (GGA) calculations, the bandgaps of SnS$_2$ and FeS$_2$ are 1.61 eV and 0.56 eV, respectively. The bandgaps of SnS$_2$ and FeS$_2$ are 2.07 eV and 0.97 eV, which are larger than those determined by the calculations. We tested a group of

simulations with different $U$ values and found that the optimized bandgap of SnS$_2$ is 2.06 eV ($U = 8.0$) and the bandgap of FeS$_2$ is 0.95 eV ($U = 1.8$), which are consistent with the experimental values[43, 44]. The calculated total density of states demonstrate that the Fe–SnS$_2$ monolayer exhibits half-metallic behavior with 100% spin-polarized carriers at the Fermi level for the down-spin channel, whereas for the up-spin channel, the monolayer exhibits a semiconducting behavior (Fig. 4b). The projected band structures of the Fe–SnS$_2$ monolayer also confirm this behavior and clearly show that the impurity levels at the Fermi level come from the Fe atom and not from the Sn atoms (Fig. 4c, f). The total magnetic moment of the Fe atom is 1.9 $\mu_B$, and the integration over all the occupied S 3$p$ states of the S atoms bonded to the Fe dopant atom yields a magnetic moment of $-0.023$ $\mu_B$. There are six S atoms around the Fe atom, leading to a total moment of $-0.14$ $\mu_B$. The distribution of spin is shown in the spin density isosurface plot in Fig. 4d. The hybridization between the localized Fe 3$d$ and the delocalized S 3$p$ states leads to an antiferromagnetic (AFM) coupling between the Fe spin and S spins. When the S spins encounter one Fe, the antiferromagnetic coupling between S and Fe leads to an effective ferromagnetic (FM) structure for all of the Fe spins. Allowing for the delocalized feature of the S 3$p$ states, the FM structure between the Fe spins is expected to emerge in a long range. The distribution of the outer electron of the Fe atom is 3$d^6$4$s^2$ based on crystal field theory, and we can obtain the distribution of 3$d$ electrons with the spin of Fe atoms in Fe–SnS$_2$ (Fig. 4e). There are three 3$d$ electrons with spin-up for the Fe atom in Fe–SnS$_2$, and one 3$d$ electron with spin-down. The magnetic moment is 2 $\mu_B$, which is consistent with the calculated value of 1.9 $\mu_B$.

The magnetic anisotropy of Fe–SnS$_2$ is due to the competition between the perpendicular and parallel spin–orbit coupling effect. The magnetic anisotropic energy (MAE) has been used to evaluate the extent of the magnetic anisotropy[45]. In this study, we assumed that the $x$ and $y$ direction are isotropic, and we took the $z$- and $x$-axes directions into account to analyze the anisotropy of

the perpendicular and parallel direction, i.e., $MAE = E(x) - E(z)$ with $E(x)$ and $E(z)$ denoting the total energy of the self-consistent calculations in the $x$ and $z$ magnetization directions, respectively[46, 47]. Our calculated value of MAE is 2.3 meV, confirming that the $z$-axis is the easy axis[48]. The theoretical result is consistent with the experimental result.

To further investigate the long-range FM ordering in Fe–$SnS_2$, a 108-atom supercell was constructed with two Fe atoms (Supplementary Fig. 14a). The magnetic energy, $\Delta E$ ($\Delta E = E_{FM} - E_{AFM}$), is −6.7 meV, showing an energetically more favorable FM coupling than an AFM coupling between the Fe spins. Supplementary Fig. 14b shows the long-range FM-ordering behavior of the Fe–$SnS_2$. This behavior has also been extensively studied via theory[25–27]. The Curie temperature, $T_C$, of the Fe–$SnS_2$ monolayer can be estimated by the relation[49]:

$$T_C = T_b / \ln\left(\frac{3\pi T_b}{4K_a}\right), \qquad (3)$$

where $T_b$ is the bulk Curie temperature, and $K_a$ is the anisotropy constant. The bulk magnetic energy, $\Delta E$, was calculated to be −14.3 meV (Supplementary Fig. 15), and the $T_b$ was estimated to be 56 K based on the mean-field theory and Heisenberg model[50]. $K_a = K_{mca} + K_{sa}$, where $K_{mca}$ is the magnetocrystalline anisotropy constant (2.3 meV), and $K_{sa}$ is the shape anisotropy constant (~−0.17 meV[49]). Thus, $K_a$ is 2.13 meV. The calculated $T_C$ is ~33 K, which is consistent with the experimental result of 31 K.

An average magnetic moment for a Fe atom is ~0.020 $\mu_B$ per atom for $Fe_{0.021}Sn_{0.979}S_2$ (Fig. 3b) and 0.021 $\mu_B$ per atom for $Fe_{0.015}Sn_{0.985}S_2$ (Supplementary Fig. 13) at 2 K. The magnetic moment of the Fe atom is relatively low[51], and this phenomenon should result from the AFM coupling between doped Fe atoms at the intralayer and interlayer sites. It has been reported that in magnetic atom-doped 2D TMDs, the difference in energies between FM and AFM coupling is related to the distances between magnetic atoms. When the distance between two magnetic atoms is long enough, it will be AFM coupling[25, 26, 29]. Wu et al. demonstrated that in Fe–$SnS_2$, Fe atoms can have AFM coupling across the layers. In the Fe–$SnS_2$ crystal[27], Fe atoms are distributed randomly (Fig. 1g), and an AFM coupling should exist and decrease the average magnetic moment of the Fe atoms.

Field-effect transistors based on this new 2D magnetic semiconductor have a high ON/OFF ratio ($>10^6$), a high electron mobility of 8.15 $cm^2 V^{-1} s^{-1}$, and a high photoresponsivity of 206 mA $W^{-1}$. Pure $SnS_2$ is diamagnetic, and Fe–$SnS_2$ shows remarkable magnetic perpendicular anisotropic behavior with a Curie temperature of 31 K. The DFT calculations confirmed the ferromagnetic behavior and the perpendicular anisotropy of Fe–$SnS_2$. The experimental and theoretical results suggest that Fe–$SnS_2$ will have excellent performance for optoelectronic devices. Magnetic atom-doped 2D materials have significant potential applications in future nanoelectronic, magnetic, and optoelectronic fields.

## Methods

**Synthesis of the Fe-doped $SnS_2$ bulk crystal.** $Fe_{0.021}Sn_{0.979}S_2$, $Fe_{0.015}Sn_{0.985}S_2$, and $Fe_{0.011}Sn_{0.989}S_2$ bulk crystals were synthesized via the direct vapor transport technique. Sn, S, and $FeCl_3$ powders were mixed in stoichiometric proportions (10:30:1, 10:30:0.8, and 10:30:0.6, respectively) and placed in an ampoule. The ampoules were pumped down to the lowest-attainable pressures in our system ($10^{-5}$ Torr) and sealed immediately to avoid blasting due to the high vapor pressure that developed inside the ampoule at growth temperatures. The ampoules were inserted into a two-zone tube furnace system, and the system was heated to 610 °C at a 30 °C $h^{-1}$ rate. Samples were left at a constant temperature for 1 day. During the growth phase, the temperature of the growth zone was gradually lowered to 600 °C at a rate of 2 °C $h^{-1}$, and the source temperature was maintained at 610 °C. After 5 days, the system was cooled down to room temperature at a rate of 10 °C $h^{-1}$.

**Characterization of the Fe-doped $SnS_2$.** Raman measurements were performed using a Renishaw micro-Raman/PL system. STEM and TEM were performed using a JEOL2100F. AFM and MFM were performed using a Bruker AFM. Magnetic measurements were performed using a Quantum design PPMS.

**DFT calculations.** According to the GGA calculations, the bandgaps of $SnS_2$ and $FeS_2$ are 1.61 eV and 0.56 eV, respectively. The bandgaps of $SnS_2$ and $FeS_2$ are 2.07 eV and 0.97 eV, which are larger than those determined via the calculations. We tested different $U$ values and found that the bandgap of $SnS_2$ is 2.06 eV ($U = 8.0$ on the $d$ orbital) and the bandgap of $FeS_2$ is 0.95 eV ($U = 1.8$ on the $d$ orbital), which are consistent with the experimental values. First-principles spin-polarized calculations were performed on the basis of DFT using project or augmented wave (PAW) potentials[52]. The exchange-correlation interactions were treated by GGA with the Perdew–Burke–Ernzerhof functional[53]. The plane-wave cutoff energy was 400 eV. Monkhorst–Pack meshes of $5 \times 5 \times 1$ and $21 \times 21 \times 1$ were employed for geometry optimization and the calculation of density of states, respectively. To obtain reliable values for the MAEs, dense $k$ points of $21 \times 21 \times 1$ were used for the calculations.

**Data availability.** The data that support the findings of this study are available from the corresponding author upon reasonable request.

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

## Acknowledgements

This work was financially supported by the National Natural Science Foundation of China (Grant No. 61622406, 11674310, 61571415, 51502283, 11574018, 11574305, and 51524701), the National Key Research and Development Program of China (Grant No. 2017YFA0207500, 2016YFB0700700, and 2016YFA0301200), the "Hundred Talents Program" of Chinese Academy of Sciences (CAS), the Strategic Priority Research Program of the CAS (Grant No. XDPB06), and the CAS/SAFEA International Partnership Program for Creative Research Teams. The authors thank Prof. Wenbo Mi (Tianjin University) and Prof. Dahai Wei (Institute of Semiconductors, CAS) for their helpful discussions.

## Author contributions

Z.W. and J.L. conceived the experiments. B.L. synthesized and characterized the single crystals. B.L. and M.Z. carried out the FET measurements. T.X. and N.L. carried out the VSM measurements. L.H. and B.L. performed ab initio calculations. J.Z. provided insightful advice. B.L. and Z.W. wrote the manuscript. All authors contributed to the scientific planning and discussions.

## Additional information

**Competing interests:** The authors declare no competing financial interests.

