## [Peer Review File · Nature Communications]

Reviewers' comments:

Reviewer #1 (Remarks to the Author):

This submission by B. Li et al. presents the idea of doping SnS₂ with iron to achieve high quality 2D nano-devices for potential optoelectronic and magnetic applications. The synthesized Fe-SnS₂ crystal is well characterized. Transport, optical and magnetic measurements are all performed. DFT calculations on monolayer samples are displayed to compare with the experiment. The result looks encouraging, but several issues need to be addressed before further consideration:

1. It is claimed in the paper that the mean free time τ of Fe doped SnS₂ monolayer should become shorter due to the enhanced carrier scattering, while the effective mass m^* of Fe-SnS₂ is slightly smaller than SnS₂. This is inconsistent with the experimental result that the mobility (equals τ/m^*) of Fe-SnS₂ monolayer is shown to be ~ 3 times larger than the SnS₂ monolayer. The mobility of monolayer SnS₂ device on SiO₂ substrate is not limited by its intrinsic scattering. Both the substrate and other disorders play an important role here. For example, using an Al₂O₃ top gate helps to enhance the field effect mobility by one order of magnitude. I believe it is hard to control all the environmental conditions to be exactly the same for SnS₂ and Fe-SnS₂ devices. One way to make the comparison more convincing is to show the mobility of the device with different Fe doping level. In this case, the enhancement (if there is any) of the mobility will be traced by at least tuning one parameter.

2. The comparison between the photoresponsivity of Fe-SnS₂ and SnS₂ is made at different experimental conditions. As the photoresponse is not linear with the laser power and is strongly dependent on the bias voltage, the author should be careful when trying to claim that the detectivity of the doped sample is more superior.

Only the photoresponsivity is discussed in detail. The speed of the photocurrent signal is not well demonstrated. From Fig. 2c, it is hard to tell the time constant of the signal and how it compares with pure SnS₂ devices.

The generation mechanism of the photoresponse is not explained, which needs to be clarified.

3. There is a lack of the magnetic characterization on monolayer Fe-SnS₂ devices. Both the VSM and the temperature dependent transport measurement are performed on the bulk crystal. I am wondering whether or not the kink in the dp/dT curve at around 40K still exists in the monolayer Fe-SnS₂.

4. Many mono-atomic layer materials are not stable in the air. It is necessary to check if there is any observable degradation of the device when it is placed in ambient.

5. Some typos and other mistakes need to be addressed, e.g. line 71 "...high-angle annular dark field" should be "...high-angle annular dark field"; Line 159 "...Magnetic hysteresis loops at room temperature for..." I believe this is actually a low temperature measurement.

Reviewer #2 (Remarks to the Author):

The authors have synthesized Fe doped SnS₂ monolayers and report good electronic properties when used in a transistor setup and large photoresponsivity. Most importantly, magnetic order with an out-of plane easy axis is found, the Curie temperature is estimated to be 31K. Density functional theory calculations have been performed to confirm these findings.

In principle, the possibility to synthesize a magnetic two-dimensional material from the

dichalcogenide class is highly attractive and the hysteresis curves in Fig.3 suggest that the process was successful. It is, however, not so clear that the experimental observations and theoretical predictions fit together.

Looking at Fig.3(b), I see that the saturation magnetization is 0.0035 emu/g. From the Fe concentration in the sample, I expect a value that would be much higher, a few emu/g, if all Fe contribute to the magnetization. How can this be explained?

I'm also a bit puzzled by the shape of the $M(T)$ curve in Fig.3(d). I can see no sign of a $(1 - T/T_C)^\beta$ behavior. How is the T_C value of 31 K obtained? It seems to be just a point in the tail of the magnetization.

Concerning the theoretical simulations, I'm also surprised: the authors cite ref. 27, where the problem of band gaps and correlation effects in dilute magnetic semiconductors is discussed and the use of DFT+U is proposed to overcome the shortcomings of density functional theory (DFT) in this case. In the present paper these problems are not even mentioned and no comparison of the exchange interaction for Fe-SnS₂ is given. Note, that in both cases the Fe concentration is much higher than in the experiment (factor 4), leading to a doubling of the average Fe-Fe distance. This should lead to a significant lowering of T_C .

It is further not clear why the authors use eq.(3) to estimate T_C , which is a formula for a three-dimensional magnet. In two dimensions, they could use the formula of Bander and Mills (Phys. Rev. B 38, 12015(R) 1988) since they have determined the anisotropy already. It will not change the result much, since K and J are of similar magnitude, but at least would be the appropriate method.

I'm also surprised that they used eq.(4) to estimate the MAE. Most modern PAW codes (the authors do not specify what they use) calculate this quantity in a self-consistent fashion. It should also be specified which unit-cell was used for this purpose and the k-point sampling density.

Finally, coming back to the experimental data, I was not really convinced by the authors' explanation of the high core-level binding energy of Fe 2p. It is true that Fe is octahedrally coordinated by S, but this also applies for FeS₂ pyrite, where the literature values for the Fe 2p level are much lower. Admittedly, pyrite is non-magnetic but the charge state should not be so different. The authors write +4, but in ref. 27 I read that the electrons lost from Fe are small, indicating more covalent character.

Summarizing, I think that the result (preparation of a magnetically doped SnS₂ monolayer with magnetic hysteresis) is nice, but wonder if it's really so clear whether the magnetization comes from the homogeneously distributed Fe. Earlier calculations found a tendency towards clustering, maybe the magnetic signal is related to that. For application, it has to be made sure that the electrons that lead to conduction are spin-polarized, to make the manuscript attractive to the readership of Nature Communications more evidence for this should be gathered.

Reviewer #3 (Remarks to the Author):

In this paper authors have reported Fe doped SnS₂ monolayer exfoliated by micromechanical cleavage technique. This work is not sufficient to warrant a publication in Nature Communications. Can be transferred to Scientific Reports.

For Reviewer 1:

Dear referee:

We appreciated your insightful suggestions which helped us to improve the quality of our manuscript. Corresponding revisions were made according to your comments.

Comments to the author:

This submission by B. Li et al. presents the idea of doping SnS₂ with iron to achieve high quality 2D nano-devices for potential optoelectronic and magnetic applications. The synthesized Fe-SnS₂ crystal is well characterized. Transport, optical and magnetic measurements are all performed. DFT calculations on monolayer samples are displayed to compare with the experiment. The result looks encouraging, but several issues need to be addressed before further consideration.

Our reply:

Thank you for your professional comments. According to your suggestions, we revised our manuscript carefully. The replies and corresponding revisions are as follows:

Comment 1. It is claimed in the paper that the mean free time τ of Fe doped SnS₂ monolayer should become shorter due to the enhanced carrier scattering, while the effective mass m^* of Fe-SnS₂ is slightly smaller than SnS₂. This is inconsistent with the experimental result that the mobility (equals τ/m^*) of Fe-SnS₂ monolayer is shown to be ~ 3 times larger than the SnS₂ monolayer. The mobility of monolayer SnS₂ device on SiO₂ substrate is not limited by its intrinsic scattering. Both the substrate and other disorders play an important role here. For example, using an Al₂O₃ top gate helps to enhance the field effect mobility by one order of magnitude. I believe it is hard to control all the environmental conditions to be exactly the same for SnS₂ and Fe-SnS₂ devices. One way to make the comparison more convincing is to

show the mobility of the device with different Fe doping level. In this case, the enhancement (if there is any) of the mobility will be traced by at least tuning one parameter.

Our reply:

Thank you for your professional comment! Mobility of 2D materials plays a critical role in the 2D materials' application in optoelectronic area in the future. Searching for high-mobility 2D materials, such as InSe and Bi₂O₂Se,^{1,2} has recently been concerned by more and more researchers.

In the analysis of mobility of Fe-SnS₂ by first principles calculations, we just consider the effective mass of Fe-SnS₂, other factors are complex and difficult to be carefully analyzed and calculated. Thus, our calculated results do not agree well with the experimental results. The mobility of 2D materials is related with its intrinsic property (such as effective mass, intrinsic scattering), scattering of substrate and electrical contact between the electrode and sample. Electrical contact is subject to the metals (different metals have different work functions), the formation of electrode (e.g., evaporation, transfer), device treatment (e.g., annealing), and so on. In order to get similar electrical contact for effective comparison, the fabrication technique and testing conditions of all the devices here are completely the same. The scattering of substrate comes from charge impurities and substrate surface roughness. In order to further investigate the enhancement of mobility in Fe-SnS₂, we use Al₂O₃/Si as the substrate to reduce the scattering of substrate. The thickness of Al₂O₃ is 30 nm and Al₂O₃ is deposited on Si substrate by atomic layer deposition (ALD). Actually, we also modulate the experimental parameters and grow three kinds of Fe-SnS₂ sample with different doping ratio (Fe_{0.021}Sn_{0.979}S₂, Fe_{0.015}Sn_{0.985}S₂ and Fe_{0.011}Sn_{0.989}S₂): Fe_{0.021}Sn_{0.979}S₂, Fe_{0.015}Sn_{0.985}S₂ and Fe_{0.011}Sn_{0.989}S₂ bulk crystals are synthesized by direct vapor transport technique. Sn, S and FeCl₃ powders are mixed in stoichiometric proportions (10:30:1, 10:30:0.8 and 10:30:0.6). The content of Fe in the samples is characterized by XPS (**Figure S5**).

Figure S5. XPS of different Fe doped SnS₂.

The conductivity (σ) without gate voltage and mobility (μ) at room temperature with different Fe content samples are listed below:

Table R1 Summary of the field-effect performance transistor parameters including electron mobility (μ), conductivity (σ) of the transistors based on our Fe-SnS₂ and SnS₂ monolayers on Al₂O₃/Si substrate.

Parameters	Fe _{0.021} Sn _{0.979} S ₂ monolayer	Fe _{0.015} Sn _{0.985} S ₂ monolayer	Fe _{0.011} Sn _{0.989} S ₂ monolayer	SnS ₂ monolayer
μ (cm ² /Vs)	8.15	6.24	4.76	3.43
σ (S/m)	15.4	11.2	7.3	4.1

The conductivity and mobility increase with the increasing of Fe content in the samples. Compared with SnS₂, Fe-SnS₂ has smaller effective mass, and thus the mobility is higher. Conductivity is related with mobility: $\sigma = n \cdot e \cdot \mu$. Where n is the carrier concentration. The results show that doping Fe atoms in SnS₂ is an effective way to enhance the mobility. In **Figures 2, S4** and **S6**, we measure the optoelectronic property of Fe-SnS₂ and SnS₂ on Al₂O₃/Si substrate. In **Figures 2** and **S4**, the measuring conditions of Fe_{0.021}Sn_{0.979}S₂ and SnS₂ are the same.

Our revision:

Figure 2. Electrical characteristics and photoresponse of $\text{Fe}_{0.021}\text{Sn}_{0.979}\text{S}_2$ monolayer. (a) Transfer and (b) Output characteristics of $\text{Fe}_{0.021}\text{Sn}_{0.979}\text{S}_2$. The inset shows the optical image of one typical device and the AFM image of the corresponding sample used for fabricating the device. (c) Time dependent I_{sd} of the transistor based on the $\text{Fe}_{0.021}\text{Sn}_{0.979}\text{S}_2$ during the light (638 nm, $2.63\ \mu\text{W}$) switching on/off under the positive source-drain voltage V_{sd} from 1 to 3 V. (d) Photoresponsivity (R) as function of light power (P) with V_{sd} of 3 V.

Figure S4. Electrical characteristics and photoresponse of SnS₂ monolayer. (a) Transfer and (b) Output characteristics of SnS₂. The inset shows the optical image of one typical device and the AFM image of the corresponding sample used for fabricating the device. (c) Time dependent I_{sd} of the transistor based on the SnS₂ during the light (638 nm, 2.63 μW) switching on/off under the positive source-drain voltage V_{sd} from 1 to 3 V. (d) Photoresponsivity (R) as function of light power (P) with V_{sd} of 3V.

Figure S6. Electrical characteristics and photoresponse of $\text{Fe}_{0.015}\text{Sn}_{0.985}\text{S}_2$ and $\text{Fe}_{0.011}\text{Sn}_{0.989}\text{S}_2$ monolayer. (a) Transfer and (b) Output characteristics of $\text{Fe}_{0.015}\text{Sn}_{0.985}\text{S}_2$. The inset shows the optical image of one typical device and the AFM image of the corresponding sample used for fabricating the device. (c) Transfer and (d) Output characteristics of $\text{Fe}_{0.011}\text{Sn}_{0.989}\text{S}_2$. The inset shows the optical image of one typical device and the AFM image of the corresponding sample used for fabricating the device.

Table S1. Summary of the field-effect performance transistor parameters including electron mobility (μ), conductivity (σ) of the transistors based on our Fe-SnS₂ and SnS₂ monolayers on Al₂O₃/Si substrate, and the previously reported SnS₂ few-layer on SiO₂/Si substrate.

Parameters	$\text{Fe}_{0.021}\text{Sn}_{0.979}\text{S}_2$ monolayer	$\text{Fe}_{0.015}\text{Sn}_{0.985}\text{S}_2$ monolayer	$\text{Fe}_{0.011}\text{Sn}_{0.989}\text{S}_2$ monolayer	SnS ₂ monolayer	SnS ₂ few-layer ^{3,4}
μ (cm ² /Vs)	8.15	6.24	4.76	3.43	5

ON/OFF ratio	7.3×10^6	6.1×10^6	6.3×10^6	1.3×10^6	4.7×10^6
R (mA/W)	206	\	\	83	8.8

Table S2. DFT calculated effective mass at conduction band bottom of SnS₂ monolayer and Fe-SnS₂ monolayer

Effective mass	$m_{M-\Gamma}^*$	m_{M-K}^*
SnS ₂ monolayer	$0.82m_0$	$0.54m_0$
Fe-SnS ₂ monolayer	$0.74m_0$	$0.47m_0$

Comment 2. The comparison between the photoresponsivity of Fe-SnS₂ and SnS₂ is made at different experimental conditions. As the photoresponse is not linear with the laser power and is strongly dependent on the bias voltage, the author should be careful when trying to claim that the detectivity of the doped sample is more superior.

Only the photoresponsivity is discussed in detail. The speed of the photocurrent signal is not well demonstrated. From Fig. 2c, it is hard to tell the time constant of the signal and how it compares with pure SnS₂ devices.

The generation mechanism of the photoresponse is not explained, which needs to be clarified.

Our reply:

Thank you for your professional comment! We agree with you that photodetectors are an important component of many optoelectronic devices, and responsivity and response time (the time constant of the signal) are the key parameters for the photodetectors.⁵ In order to better compare the photoresponsivity of Fe-SnS₂ and SnS₂, we have re-measured the devices (the substrate is Al₂O₃/Si) and made all experimental conditions completely the same for precise comparison. In general, there are three mechanisms enabling photodetection: photoconduction, photogating and the photovoltaic effect. The transfer characteristic of Fe_{0.021}Sn_{0.979}S₂ under illumination, different from the dark one, shows an increasing in the conductivity (vertical shift) and a positive photocurrent across the entire gate voltage range (**Figure R1**). The

result exhibits that the mechanism of photodetection is photoconduction. The schematic of photocurrent generation is clarified in **Figure S8**. The absorption of photons generates electron-hole pairs that are separated by the external applied bias, generating a photocurrent and reducing the electrical resistance of the semiconductor. The photocurrent (I_p) can be estimated with the following formula:⁶

$$I_p = \Gamma \cdot \eta \cdot e \frac{\tau_p \cdot \mu \cdot V}{L^2} \quad (1)$$

Where Γ is the number of absorbed photons per unit time, η is the efficiency of the conversion of the absorbed photons to electrons, e is the electron charge, τ_p is photogenerated carrier lifetime, μ is the mobility and V is the source-drain bias, L is the length of the transistor channel. The mobility of Fe-SnS₂ is larger than that of SnS₂ for the smaller effective mass, and the photogenerated carrier lifetime of Fe-SnS₂ should be smaller than that of SnS₂ for the impurity levels that promoting the electron-hole recombination. The response time is strongly related with the photogenerated carrier lifetime and the response time of Fe-SnS₂ (9 ms) is larger than that of SnS₂ (6 ms) under the same condition (**Figure S7**). On the whole, the photoresponsivity of Fe-SnS₂ is larger than that of SnS₂ under the same laser power, and the response time of Fe-SnS₂ is larger than that of SnS₂. As photoresponsivity and response time are two important parameters to evaluate the photodetector, it is incorrect to claim that the detectivity of the Fe-SnS₂ is superior. We have revised the statement in the paper. In the previous version, we claimed “The results suggest that Fe-SnS₂ has the **enhanced** performance of optoelectronic devices and its curie temperature is 31 K, thus open up new realms for future nanoelectronic, magnetic, and optoelectronic applications.” In the revised version, we claim “The results suggest that Fe-SnS₂ has the **excellent** performance of optoelectronic devices and its curie temperature is 31 K, thus open up new realms for future nanoelectronic, magnetic, and optoelectronic applications.”

Figure R1. Transfer characteristics of $\text{Fe}_{0.021}\text{Sn}_{0.979}\text{S}_2$ under dark and illumination.

Figure S8. Schematic of the photoconductive effect for SnS_2 and Fe-SnS_2 . (a) Band alignment for SnS_2 contacted with two metals (Au) under an external bias without illumination. (b) Band alignment under illumination for SnS_2 . (c) Band alignment for Fe-SnS_2 contacted with two metals (Au) under an external bias without illumination. (d) Band alignment under illumination for Fe-SnS_2 .

Figure S7. High-resolution time response of (a) SnS₂ and (b) Fe-SnS₂ monolayer measured with V_{sd} of 1 V and light power of 2.63 μW .

Our revision:

Based on the transport curve (**Figure 2a**), the calculated electron mobility is 8.15 cm^2/Vs . Fe_{0.015}Sn_{0.985}S₂ and Fe_{0.011}Sn_{0.989}S₂ have also been synthesized by modulating the growth condition (**Figure S5**). The electronic transport property of Fe_{0.015}Sn_{0.985}S₂ and Fe_{0.011}Sn_{0.989}S₂ (**Figure S6**) have been investigated and the calculated mobility are larger than the values of pure SnS₂ (**Table S1**). The mobility increases with the increasing of Fe content in the samples.

Figure S7. High-resolution time response of (a) SnS₂ and (b) Fe-SnS₂ monolayer measured with V_{sd} of 1 V and light power of 2.63 μW .

Discussion of optoelectronic property of Fe_{0.021}Sn_{0.979}S₂ monolayer

The mechanism of photocurrent generation is clarified in **Figure S6**. The absorption

of photons generates electron-hole pairs that are separated by the external applied bias, generating a photocurrent and reducing the electrical resistance of the semiconductor. The photocurrent (I_p) can be estimated with the following formula:

$$I_p = \Gamma \cdot \eta \cdot e \frac{\tau_p \cdot \mu \cdot V}{L^2}$$

Where Γ is the number of absorbed photons per unit time, η is the efficiency of the conversion of the absorbed photons to electrons, e is the electron charge, τ_p is photogenerated carrier lifetime, μ is the mobility and V is the source-drain bias, L is the length of the transistor channel. The mobility of Fe-SnS₂ is larger than that of SnS₂ for the smaller effective mass, and the photogenerated carrier lifetime of Fe-SnS₂ should be smaller than that of SnS₂ for the impurity levels that promoting the electron-hole recombination. The response time is strongly related with the photogenerated carrier lifetime and the response time of Fe-SnS₂ is larger than that of SnS₂ under the same condition (**Figure S7**).

Figure S8. Schematic of the photoconductive effect for SnS₂ and Fe-SnS₂. (a) Band alignment for SnS₂ contacted with two metals (Au) under an external bias without illumination. (b) Band alignment under illumination for SnS₂. (c) Band alignment for Fe-SnS₂ contacted with two metals (Au) under an external bias without illumination. (d) Band alignment under illumination for Fe-SnS₂.

Comment 3. There is a lack of the magnetic characterization on monolayer Fe-SnS₂ devices. Both the VSM and the temperature dependent transport measurement are performed on the bulk crystal. I am wondering whether or not the kink in the dp/dT curve at around 40K still exists in the monolayer Fe-SnS₂.

Our reply:

We agree with you that magnetic characterization on 2D monolayer materials will be very interesting. 2D magnetic materials are attracting great attentions recently, Zhang *et al.* (*Nature* 2017, **546**, 265-269, published online April 26th, 2017) and Xu *et al.* (*Nature* 2017, **546**, 270-273, published online June 7th, 2017) reported the magnetic property of Cr₂Ge₂Te₆ and CrI₃ monolayer by high-resolution and high-sensitivity magneto-optic microscopy.^{7,8} These two systems are intrinsic magnetic materials with layered structure. **But we used a completely different method (doping) to achieve the 2D ferromagnetic materials based on the intrinsic diamagnetic material (SnS₂).** In our Fe-SnS₂, Fe atoms are doped at the sites of Sn atoms randomly with very low concentration (2.1%), therefore the magnetic signal here is very weak. The detecting limitation of VSM is about 10⁻⁷ emu, and the mass of our Fe-SnS₂ bulk crystal sheet used for VSM measurement is about 1 mg. The magnetic signal is on the order of 10⁻⁶ emu as shown in **Figure 3**, closing to the detecting limitation of VSM. It is roughly evaluated that the magnetic signal should be less than 10⁻¹⁰ emu for Fe-SnS₂ monolayer, beyond the detecting limitation of VSM system. And it also is a big challenge for us to get a very large scale monolayer sample by mechanical exfoliation. Thus, we failed to achieve the VSM measurement of Fe-SnS₂ monolayer. We have also try to use the Physical Properties Measurement System (PPMS) to measure the spin transport in Fe-SnS₂. Unfortunately, the lowest detecting limitation of resistivity is about 10⁻¹ MΩ for PPMS, but the resistivity of Fe-SnS₂ is about 10⁻² MΩ. Thus we did not get good result (**Figure R2**). We appreciate for your understanding of our challenge on the magnetic characterization on monolayer, and due to the time limitation of this revision, we will try to carry out more other methods

in future study.

Figure R2. (a) Equipment photographs of Physical Properties Measurement System (PPMS). (b) Optical image of the $\text{Fe}_{0.021}\text{Sn}_{0.979}\text{S}_2$ few-layer device used for spin transport measurement.

Following your comment about the $d\rho/dT$ curve of monolayer Fe-SnS_2 , during this revision we have measured temperature dependent transport property of $\text{Fe}_{0.021}\text{Sn}_{0.979}\text{S}_2$ monolayer and also thicker samples from 200 to 8 K. It is observed that the temperature derivative of the measured resistivity of the monolayer has a kink at about 32 K, a little bit different from 41 K of 13 and 35 nm thick samples (**Figure S10**). The $d\rho/dT$ curves of samples with different thickness have similar behavior, and they all have kink near Curie temperature (31 K) of bulk crystal sheet. It is suggested these similar trend should come from the magnetic transition, and the monolayer should have same physical properties including the magnetism as bulk crystals. As the kink location is related with the thickness, the coupling between layers may have impact on the magnetism.

Figure S10. Resistivity and temperature derivative of the measured resistivity of Fe-SnS₂ nanosheets with the thickness of monolayer, 13 and 35 nm.

Our revision:

The resistivity of the Fe_{0.021}Sn_{0.979}S₂ thin film with different thickness has been measured from 200 K to 8 K (**Figure S10**). It is observed that the temperature derivative of the measured resistivity have a transition at about 32 K of monolayer and 41 K of 13 and 35 nm thick samples, which are close to the Curie temperature of 31 K. It is suggested that this resistive transition could come from the magnetic transition. We will discuss the origin of the magnetism and the magnetic anisotropy of Fe_{0.021}Sn_{0.979}S₂ sheet in detail.

Figure S10. Resistivity and temperature derivative of the measured resistivity of Fe-SnS₂ nanosheets with the thickness of monolayer, 13 and 35 nm.

Comment 4. Many mono-atomic layer materials are not stable in the air. It is necessary to check if there is any observable degradation of the device when it is placed in ambient.

Our reply:

Thank you for your professional comment! Stability of 2D atomic layer materials is very critical for their real application in future, both for the fundamental research and also industrial realization. Here, we have measured the electrical transport property of one Fe_{0.021}Sn_{0.979}S₂ monolayer FETs every three days for a month. The transfer and output curves are shown in **Figure R3**. The storage and measurement for the device were all carried in atmosphere (during this period, the room temperature is 18-31 °C, and the humidity is 30-70% in our lab at Beijing, China.), and the storage was also

maintained without absent of light. After one month, the mobility changes from $6.1 \text{ cm}^2\text{V}^{-1}\text{s}^{-1}$ to $4.7 \text{ cm}^2\text{V}^{-1}\text{s}^{-1}$, and the ON/OFF ratio changes from 1.2×10^6 to 7×10^5 (**Figure S9**). The results show that the $\text{Fe}_{0.021}\text{Sn}_{0.979}\text{S}_2$ is very stable and has large potential application in optoelectronic. For their future utilization under versatile environments, we think the normal encapsulation can further increase the stability of our devices based on Fe-SnS₂.

Figure R3. Electrical stability of $\text{Fe}_{0.021}\text{Sn}_{0.979}\text{S}_2$ monolayer device. (a) Optical image of the FETs. (b)-(i) The transfer and output curves of the $\text{Fe}_{0.021}\text{Sn}_{0.979}\text{S}_2$ monolayer FETs measuring in the air in 0 to 30 days, respectively.

Figure S9. ON/OFF ratio and mobility of a typical $\text{Fe}_{0.021}\text{Sn}_{0.979}\text{S}_2$ monolayer FET measured for one month.

Our revision:

Stability of 2D atomic layer materials is critical for their application in future. Electrical property of a typical $\text{Fe}_{0.021}\text{Sn}_{0.979}\text{S}_2$ monolayer FETs on SiO_2/Si substrate stored in air is measured for one month. After one month, the mobility changes from $6.1 \text{ cm}^2\text{V}^{-1}\text{s}^{-1}$ to $4.7 \text{ cm}^2\text{V}^{-1}\text{s}^{-1}$, and the ON/OFF ratio changes from 1.2×10^6 to 7×10^5 (**Figure S9**). The results show that the $\text{Fe}_{0.021}\text{Sn}_{0.979}\text{S}_2$ is very stable and has large potential application in optoelectronic area.

Figure S9. ON/OFF ratio and mobility of a typical $\text{Fe}_{0.021}\text{Sn}_{0.979}\text{S}_2$ monolayer FET measured for one month.

Comment 5. Some typos and other mistakes need to be addressed, e.g. line 71 “...high-angle annular dark filed” should be “...high-angle annular dark field”; Line 159 “...Magnetic hysteresis loops at room temperature for...” I believe this is actually a low temperature measurement.

Our reply:

Thank you for your kind reminder! We have checked the whole paper carefully and revised all typos and mistakes. Line 159 should be “...Magnetic hysteresis loops at 2 K for...”

Our revision:

The crystallinity of the $\text{Fe}_{0.021}\text{Sn}_{0.979}\text{S}_2$ is further characterized using high-angle annular dark field scanning transmission electron microscopy (HAADF-STEM) and transmission electron microscopy (TEM).

The measurements are carried out in two types of applied magnetic field (H): perpendicular to the sheet, e.g. parallel to the [001] direction (H_{\perp}), and parallel to the sheet, e.g. perpendicular to the [001] direction (H_{\parallel}). Magnetic hysteresis loops at 2 K for SnS_2 and $\text{Fe}_{0.021}\text{Sn}_{0.979}\text{S}_2$ have been shown in **Figures 3a and b**.

For Reviewer 2:

Dear referee:

We appreciated your professional suggestions which helped us to improve the quality of our manuscript. Corresponding revisions were made according to your comments.

Comments to the author:

The authors have synthesized Fe doped SnS₂ monolayers and report good electronic properties when used in a transistor setup and large photoresponsivity. Most importantly, magnetic order with an out-of plane easy axis is found, the Curie temperature is estimated to be 31K. Density functional theory calculations have been performed to confirm these findings.

In principle, the possibility to synthesize a magnetic two-dimensional material from the dichalcogenide class is highly attractive and the hysteresis curves in Fig.3 suggest that the process was successful. It is, however, not so clear that the experimental observations and theoretical predictions fit together.

Our reply:

Thank you for your professional comments. According to your suggestions, we revised our manuscript carefully. The replies and corresponding revisions are as follows:

Comment 1. Looking at Fig.3(b), I see that the saturation magnetization is 0.0035 emu/g. From the Fe concentration in the sample, I expect a value that would be much higher, a few emu/g, if all Fe contribute to the magnetization. How can this be explained?

Our reply:

This is a good question. The calculated magnetic moment of single Fe atom is $2 \mu_B$ by first-principles calculation. In experiment, based on **Figure 3b**, we obtain the magnetic moment of Fe atom $\sim 0.02 \mu_B/\text{atom}$. The decreasing of the experimental values has also been reported in other diluted magnetic materials, such as Mn doped ZnO and Co doped TiO_2 .^{9,10} In the Mn doped ZnO, the calculated magnetic moment of Mn is $5 \mu_B/\text{atom}$, and the experimental value is $0.16 \mu_B/\text{atom}$.⁹ The decreasing of the magnetic moment in experiment should come from the complex interaction of the sample. In theory, it is reported that in the magnetic atom doped 2D TMDs, the difference in energies between FM and AFM coupling is related with the distances of magnetic atoms. When the distance between two magnetic atoms is long enough, it will be AFM coupling,¹¹⁻¹³ and these atoms have no effect on the total FM magnetic moment. In addition, there are large and inevitable defects (e.g. vacancy) in the sample, which will also impact the magnetic moment. In theory, what we consider is the perfect situation. Thus, in Fe-SnS₂ crystal, the experimental magnetic moment is smaller than the calculated one.

Comment 2. I'm also a bit puzzled by the shape of the M(T) curve in Fig.3(d). I can see no sign of a $(1-T/T_C)^\beta$ behavior. How is the T_C value of 31 K obtained? It seems to be just a point in the tail of the magnetization.

Our reply:

We agree with you that it is a good way to obtain the Curie temperature by Curie-Weiss law. The Curie-Weiss law describes the magnetic susceptibility χ of ferromagnetic material in the paramagnetic region above the Curie point:

$$\chi = \frac{C}{T-T_C} \quad (2)$$

In many materials the Curie-Weiss law fails to describe the susceptibility in the immediate vicinity of the Curie point, since it is based on a mean-field approximation. Instead, there is a critical behavior of the form:

$$\chi = \frac{C}{(T-T_C)^\beta} \quad (3)$$

Here, β is the critical exponent. It is precise and valid to find Curie temperature by this equation. In our M-T curve shown in **Figure 3d**, the signal is weak and the data is difficult to be precisely fit to curves with this equation. We just roughly obtain the Curie temperature from the inflection point at about 31 K. In order to obtain the Curie temperature more effectively, Magnetic hysteresis loops of the $\text{Fe}_{0.021}\text{Sn}_{0.979}\text{S}_2$ bulk crystal are acquired under different temperature (**Figure S11**). It is obviously observed that the sample is ferromagnetic at 25 K and 30 K, and is paramagnetic at 35 K and 40 K. The Curie temperature should be between 30 and 35 K, which is consistent with the M-T curve in **Figure 3d**. Thus, it is clarified that the Curie temperature is about 31 K.

Figure S11. Magnetic hysteresis loops for $\text{Fe}_{0.021}\text{Sn}_{0.979}\text{S}_2$ bulk at 40 K, 35 K, 30 K and 25 K using VSM, respectively.

Our revision:

Magnetism versus temperature curves shows that the Curie temperature T_C is about 31 K (**Figure 3d**). Magnetic hysteresis loops of the $\text{Fe}_{0.021}\text{Sn}_{0.979}\text{S}_2$ bulk crystal are acquired under different temperature (**Figure S11**), and the result exhibits that the Curie temperature is between 35 K and 30 K, consistent with magnetism versus temperature curves (**Figure 3d**).

Figure S11. Magnetic hysteresis loops for $\text{Fe}_{0.021}\text{Sn}_{0.979}\text{S}_2$ bulk at 40 K, 35 K, 30 K and 25 K using VSM, respectively.

Comment 3. Concerning the theoretical simulations, I'm also surprised: the authors cite ref. 27, where the problem of band gaps and correlation effects in dilute magnetic semiconductors is discussed and the use of DFT+U is proposed to overcome the shortcomings of density functional theory (DFT) in this case. In the present paper these problems are not even mentioned and no comparison of the exchange interaction for Fe-SnS_2 is given. Note, that in both cases the Fe concentration is much higher than in the experiment (factor 4), leading to a doubling of the average Fe-Fe distance. This should lead to a significant lowering of T_C .

Our reply:

We agree with you that the bandgaps of materials with transition atoms calculated by GGA are always lower than that in experiment. By the GGA calculation, the bandgaps of SnS_2 and FeS_2 are 1.61 eV and 0.56 eV, respectively. The bandgaps of SnS_2 and FeS_2 are 2.07 eV and 0.97 eV, which are larger than those in theory. Here, we test different U value (**Figure R4**) and find that $U=8.0$ for SnS_2 and $U=1.8$ for FeS_2 consistent with the experimental value (**Tables R2** and **R3**). In order to make the VASP calculation better conform to the experiment, we construct 108-atom supercell structure of Fe-SnS_2 with 2 Fe atoms. The magnetic energy ΔE ($\Delta E=E_{\text{FM}}-E_{\text{AFM}}$) is -6.7 meV, showing that ferromagnetic coupling between Fe atoms is stable. ΔE of

108-atom supercell with 2 Fe atoms is smaller than that of 75-atom supercell with 2 Fe atoms calculated before ($\Delta E = -9.7$ meV). The Curie temperature can be estimated by two methods:

$$k_B T_C = \frac{2}{3} \Delta E \quad (4)$$

$$T_C = T_b / \ln\left(\frac{3\pi T_b}{4K_a}\right) \quad (5)$$

The Curie temperature T_C is calculated to be 37.9 K and 34.7 K using equation (4) and (5), respectively, consisting with the experimental result of about 31 K.

In **Figures 4** and **S13**, we calculated 27-atom Fe-SnS₂ with one Fe atom and 108-atom Fe-SnS₂ with two Fe atoms by DFT+U.

Table R2 Bandgaps vary with U values for SnS₂ monolayer.

U	0	1	2	3	4	5	6	7	8	9	10	11
Bandgap (eV)	1.61	1.66	1.72	1.78	1.81	1.86	1.93	1.97	2.06	2.09	2.10	2.12

Table R3 Bandgaps vary with U values for FeS₂.

U	0	0.3	0.6	0.9	1.2	1.5	1.8	2.1	2.4	2.7	3.0
Bandgap (eV)	0.56	0.62	0.69	0.76	0.81	0.88	0.95	1.14	1.21	1.27	1.31

Figure R4. DFT+U calculations of SnS₂, the U values vary from 0 to 11.

Figure S13. (a) 108-atom supercell structure of Fe-SnS₂. (b) The corresponding isosurface plots showing the spin charge density. The isosurface value is taken at $0.001 \text{ e}/\text{\AA}^3$.

Our revision:

By the GGA calculation, the bandgaps of SnS₂ and FeS₂ are 1.61 eV and 0.56 eV, respectively. The bandgaps of SnS₂ and FeS₂ are 2.07 eV and 0.97 eV, which are larger than those in theory. Here, we test different U value and find that U=8.0 for SnS₂ and U=1.8 for FeS₂ consistent with the experimental value.

Figure 4. Theoretical calculations of Fe-SnS₂ monolayer. (a) Atomic structure of Fe-SnS₂ monolayer. Red, yellow and blue balls stand for Sn, S and Fe atoms. (b) Total density of states of Fe-SnS₂ monolayer. (c). (f) Projected band structures of Fe-SnS₂ monolayer for up-spin and down-spin channels, respectively. Red and blue circles denote the contribution of Sn and Fe atoms in the total band structure, respectively. The Fermi level is set as zero. (d) The distribution of the spin density in real space for Fe-SnS₂ monolayer. The isosurface value is taken at 0.001 e/Å³. (e) The schematic of Fe 3d electrons arrangement with spin in Fe-SnS₂.

Figure S13. (a) 108-atom supercell structure of Fe-SnS₂. (b) The corresponding

isosurface plots showing the spin charge density. The isosurface value is taken at $0.001 \text{ e}/\text{\AA}^3$.

Comment 4. It is further not clear why the authors use eq.(3) to estimate T_C , which is a formula for a three-dimensional magnet. In two dimensions, they could use the formula of Bander and Mills (Phys. Rev. B 38, 12015(R) 1988) since they have determined the anisotropy already. It will not change the result much, since K and J are of similar magnitude, but at least would be the appropriate method.

Our reply:

We use the equation (3) ($k_B T_C = 2\Delta E/3$) in the previous version of the paper, because U. Schwingenschlöggl et al. used this equation to estimate the Curie temperature in magnetic atoms doped MoS_2 .¹¹ Considering the two-dimensional structure and anisotropy, we agree with you and adopt the equation (16) in the work of Bander and Mills (Phys. Rev. B 38, 12015(R) 1988) to estimate the Curie temperature:

$$T_C = T_b / \ln\left(\frac{3\pi T_b}{4K_a}\right) \quad (6)$$

Where T_b is the bulk Curie temperature, K_a is the anisotropy constant. T_b is 187 K. $2K_a = 4\pi M_s d_c$, d_c is the number of layers and $-4\pi M_s$ is related with the dipolar anisotropy. $4\pi M_s \approx 2 \times 10^4 \text{ G}$ and $d_c = 1$. Then $K \approx 1 \times 10^4 \text{ G} \approx 2 \text{ K}$. The estimated T_C is about 34.7 K, consisting with the experiment result of 31 K.

Our revision:

The Curie temperature T_C of the Fe-SnS_2 monolayer can be estimated by the relation:⁴⁴

$$T_C = T_b / \ln\left(\frac{3\pi T_b}{4K_a}\right) \quad (3)$$

Where T_b is the bulk Curie temperature, K_a is the anisotropy constant. T_b is 187 K,²⁷ and K_a is about 2 K.⁴⁴ The calculated T_C is about 34.7 K, consistent with the experiment result of 31 K.

Comment 5. I'm also surprised that they used eq.(4) to estimate the MAE. Most modern PAW codes (the authors do not specify what they use) calculate this quantity in a self-consistent fashion. It should also be specified which unit-cell was used for this purpose and the k-point sampling density.

Our reply:

In the previous version, some details of calculation were not listed. After discussion with Prof. Wenbo Mi (Tianjin University, he is good at magnetic anisotropy.^{14,15}), we have a better understanding of magnetic anisotropy. To obtain reliable values of MAE, the Gaussian smearing method with a smaller smearing of 0.05 and dense k points of $21 \times 21 \times 1$ were used in noncollinear calculations, where the SOC term was included using the second-variation method employing the scalar-relativistic eigenfunctions of the valence states. The unit-cell uFSSed for calculation is 27 atoms with 1 Fe atom in

Figure 4a.

Figure 4. Theoretical calculations of Fe-SnS₂ monolayer. (a) Atomic structure of Fe-SnS₂ monolayer. Red, yellow and blue balls stand for Sn, S and Fe atoms. (b) Total density of states of Fe-SnS₂ monolayer. (c). (f) Projected band structures of Fe-SnS₂ monolayer for up-spin and down-spin channels, respectively. Red and blue circles denote the contribution of Sn and Fe atoms in the total band structure, respectively. The Fermi level is set as zero. (d) The distribution of the spin density in

real space for Fe-SnS₂ monolayer. The isosurface value is taken at 0.001 e/Å³. (e) The schematic of Fe 3d electrons arrangement with spin in Fe-SnS₂.

Our revision:

First-principles spin-polarized calculations are performed on the basis of density functional theory (DFT) using projector-augmented wave (PAW) potentials.³ The exchange-correlation interactions are treated by the generalized gradient approximation (GGA) with Perdew–Burke–Ernzerhof (PBE) functional.⁴ The plane-wave cutoff energy is 400 eV. Monkhorst–Pack (MP) meshes of 5×5×1 and 21×21×1 are employed for geometry optimization and calculation of density of states, respectively. To obtain reliable values of MAE, the Gaussian smearing method with a smaller smearing of 0.05 and dense k points of 21 × 21 × 1 were used in noncollinear calculations, where the SOC term was included using the second-variation method employing the scalar-relativistic eigenfunctions of the valence states.

Comment 6. Finally, coming back to the experimental data, I was not really convinced by the authors' explanation of the high core-level binding energy of Fe 2p. It is true that Fe is octahedrally coordinated by S, but this also applies for FeS₂ pyrite, where the literature values for the Fe 2p level are much lower. Admittedly, pyrite is non-magnetic but the charge state should not be so different. The authors write +4, but in ref. 27 I read that the electrons lost from Fe are small, indicating more covalent character.

Our reply:

This is a very good question. It is observed that in Fe-SnS₂ few layers, Fe atoms are doped at the sites of Sn atoms (**Figure 1h**). It is true that Fe atoms in Fe-SnS₂ and FeS₂ are octahedrally coordinated by S. The distribution of outer electron of Fe atom is 3d⁶4s². **FeS₂ is non-magnetic, the valance of Fe is +2 and S is -1.**¹⁶ In SnS₂, the valance of Sn is +4 and S is -2. Based on the crystal field theory (3d⁶ electrons in octahedral coordination), **Figure R5b** shows the distribution of six electron of Fe²⁺ in

FeS₂. Obviously, the magnetic moment of Fe²⁺ is zero, and FeS₂ shows non-magnetic. Different from FeS₂, the magnetic moment of Fe atom in Fe-SnS₂ is 2μ_B, which can be calculated by VASP.¹⁷ The total DOS and band structure are shown in **Figures 4b, c and f**, based on the crystal field theory, we can obtain the distribution of 3d electrons with spin of Fe atom in Fe-SnS₂. There are three 3d electrons with spin-up of Fe atom in Fe-SnS₂, and one 3d electron with spin-down (**Figure 4e**). Thus the magnetic moment is 2μ_B (3-1=2), consistent with the calculated result. In this case, Fe in Fe-SnS₂ is 4+, because two 4s electrons and two 3d electrons are lost. In addition, XPS peak of Fe 3p_{3/2} in FeS₂ is 707 eV,¹⁸ which is lower than that of Fe-SnS₂ (**Figure S3**). As the coordination of Fe atoms in Fe-SnS₂ and FeS₂ are the same, this should be the powerful proof that the valence of Fe atoms in Fe-SnS₂ is higher than that in FeS₂. On the whole, the coordination of Fe atoms in Fe-SnS₂ and FeS₂ are the same, but the valence of S is different, and the magnetic moment and valence of Fe are different.

Figure R5. (a) Atomic structure of FeS₂. (b) The schematic of Fe 3d electrons arrangement with spin in FeS₂.

Figure 4. Theoretical calculations of Fe-SnS₂ monolayer. (a) Atomic structure of Fe-SnS₂ monolayer. Red, yellow and blue balls stand for Sn, S and Fe atoms. (b) Total density of states of Fe-SnS₂ monolayer. (c). (f) Projected band structures of Fe-SnS₂ monolayer for up-spin and down-spin channels, respectively. Red and blue circles denote the contribution of Sn and Fe atoms in the total band structure, respectively. The Fermi level is set as zero. (d) The distribution of the spin density in real space for Fe-SnS₂ monolayer. The isosurface value is taken at $0.001 \text{ e}/\text{\AA}^3$. (e) The schematic of Fe 3d electrons arrangement with spin in Fe-SnS₂.

Our revision:

The distribution of outer electron of Fe atom is $3d^64s^2$, based on the crystal field theory, we can obtain the distribution of 3d electrons with spin of Fe atom in Fe-SnS₂ (Figure 4e). There are three 3d electrons with spin-up of Fe atom in Fe-SnS₂, and one 3d electron with spin-down. The magnetic moment is $2 \mu_B$, consistent with the calculated of $1.9 \mu_B$.

Figure 4. Theoretical calculations of Fe-SnS₂ monolayer. (a) Atomic structure of Fe-SnS₂ monolayer. Red, yellow and blue balls stand for Sn, S and Fe atoms. (b) Total density of states of Fe-SnS₂ monolayer. (c), (f) Projected band structures of Fe-SnS₂ monolayer for up-spin and down-spin channels, respectively. Red and blue circles denote the contribution of Sn and Fe atoms in the total band structure, respectively. The Fermi level is set as zero. (d) The distribution of the spin density in real space for Fe-SnS₂ monolayer. The isosurface value is taken at $0.001 \text{ e}/\text{\AA}^3$. (e) The schematic of Fe 3d electrons arrangement with spin in Fe-SnS₂.

Comment 7. Summarizing, I think that the result (preparation of a magnetically doped SnS₂ monolayer with magnetic hysteresis) is nice, but wonder if it's really so clear whether the magnetization comes from the homogeneously distributed Fe. Earlier calculations found a tendency towards clustering, maybe the magnetic signal is related to that. For application, it has to be made sure that the electrons that lead to conduction are spin-polarized, to make the manuscript attractive to the readership of Nature Communications more evidence for this should be gathered.

Our reply:

We agree with you that cluster is easy formed in the doped materials. It is found that Fe cluster can be existed when the content of initial Fe source increases (**Figure S12**). In the Fe-SnS₂ with Fe cluster, there will be additional Fe 2p peak in XPS and the

Curie temperature is about 368 K. That is to say, if there are Fe cluster in the Fe-SnS₂ sample, the magnetic property should be very different. In addition, AFM (**Figure 1b**), Raman spectra (**Figure 1c**), STEM (**Figure 1g**) and XPS (**Figure S3**) of Fe_{0.021}Sn_{0.979}S₂ show no Fe cluster in the sample. XRD patterns of Fe_{0.021}Sn_{0.979}S₂ and pure SnS₂ agree well with each other and no additional peak is existed in the Fe_{0.021}Sn_{0.979}S₂ (**Figure R6**). The XRD peaks of Fe_{0.021}Sn_{0.979}S₂ are slightly broader than that of SnS₂, which should be induced by doped Fe atoms. The result has further confirmed that no Fe clusters in the Fe_{0.021}Sn_{0.979}S₂.

Spin transport in semiconductor, especially 2D semiconductor, is attracting particular attention for the great potential application in spintronics. Recently, Zhang et al. and Xu et al. characterized the magnetic property of Cr₂Ge₂Te₆ and CrI₃ monolayer by high-resolution and high-sensitivity magneto-optic microscopy.^{7,8} Spin transport in 2D materials is interesting, but there are great challenges. In this period, we are trying to measure the spin transport in Fe-SnS₂, but the experiment is not proceeding well. The challenges are: Firstly, electrical contact between the electrode and sample is a problem and needs further optimization.¹⁹ Secondly, substrate scattering will have a big effect on the spin transport. Lastly, the fabrication of spin-transport device need high technology for the spin diffusion length may be less than 1 μm.¹⁹ We would like to further cooperate with others to investigate the spin transport of 2D Fe-SnS₂.

Figure S3. XPS spectra of the exfoliated Fe-SnS₂ flakes on the substrate. XPS spectra of full region (a), Sn 3d peaks (b), S 2p peaks (c) and Sn 3p, Fe 2p peaks (d).

Figure S12. (a) XPS spectra of Fe-SnS₂ with Fe cluster. (b) Magnetic hysteresis loops Fe-SnS₂ with Fe cluster at 300 K using VSM. (c) The expanded view of the loop in (b). (d) Magnetization as a function of temperature for the Fe-SnS₂ with Fe cluster from 300 to 450 K.

Figure R6. XRD patterns of SnS₂ and Fe-SnS₂.

For Reviewer 3:

Dear referee:

Thank you very much for your evaluation on our work. During this revision, in order to improve the quality of our manuscript, we have made great efforts with adding groups of new experimental results and theoretical modeling to get more effective evidences and discussions. We hope it can reach the high request of Nature Communications now and appreciate your reconsideration about our manuscript.

Comments to the author:

In this paper authors have reported Fe doped SnS₂ monolayer exfoliated by micromechanical cleavage technique. This work is not sufficient to warrant a publication in Nature Communications. Can be transferred to Scientific Reports.

Our reply:

Thank you for your comments. In this paper, we reported high-quality Fe doped SnS₂ monolayer exfoliated by micromechanical cleavage method. After the major revision based on the comments of the reviewers, we have carried groups of new experiments. We think the characterization is more comprehensive and the discussion is deeper now, and the quality of this paper has a great improvement. The highlights of this paper are listed below:

(1) 2D van der Waals materials have great potential application in the integrated circuit for the atomic layer, and our work showed that doping is an effective method to achieve magnetic low-dimensional semiconductors and the Fe-SnS₂ is an exciting/promising candidate in the future magnetic storage.

2D magnetic materials are attracting particular attention for the great potential application in spintronics and optoelectronics. During our submission to *Nat. Commun.* (our paper was submitted on March 28th, 2017), Zhang *et al.* and Xu *et al.* reported the magnetic property of 2D Cr₂Ge₂Te₆ (Zhang *et al. Nature* 2017, **546**, 265-269, published online April 26th, 2017) and CrI₃ (Xu *et al. Nature* 2017, **546**,

270-273, published online June 7th, 2017) van der Waals crystals by high-resolution and high-sensitivity magneto-optic microscopy.^{7,8} These two great works reveal the fantastic and bright potential of 2D ferromagnetic atomic crystals in the future electronics. Different from these two intrinsic magnetic materials, our Fe-SnS₂ sample shows doping induce ferromagnetic based on the intrinsic diamagnetic material (SnS₂). The content of Fe can be intentionally tuned by modulating the growth condition (**Figure S5**). Herein, magnetic behavior of Fe-SnS₂ shows obviously perpendicular magnetic anisotropy (PMA) which has attracted much attention because of its potential applications in magnetic random access memories (MRAM) (**Figure R7**). Our work may provide some other opportunities and abundant physical pictures in further studying 2D magnetic materials by the magnetic doping of versatile 2D family.

Figure S5. XPS of different Fe doped SnS₂.

Figure R7. Distribution of the spin density in real space for Fe-SnS₂ monolayer, and magnetic hysteresis loops for Fe_{0.021}Sn_{0.979}S₂ at 2 K using VSM.

(2) The good stability of Fe-SnS₂ could obviously expand its future application and cross research, such as constructing different van der Waals heterostructures, measuring magneto electrical property.

Stability of 2D atomic layer materials is critical for their application in future. Different from Cr₂Ge₂Te₆ and CrI₃, our Fe-SnS₂ showed good stability in the air and the property of field-effect transistors decay slowly as time goes on (**Figure S9**). The storage and measurement for the device were all carried in atmosphere (during this period, the room temperature is 18-31 °C, and the humidity is 30-70% in our lab at Beijing, China.), and the storage was also maintained without light.

Figure S9. ON/OFF ratio and mobility of one Fe_{0.021}Sn_{0.979}S₂ monolayer FETs measured for one month.

(3) Our results also showed that doping is also an efficient method to improve the (opto)electronic performance of 2D semiconductors.

There are numerous reports about the doping or alloying of 2D atomic crystals^{20,21} with the purpose of band gap engineering towards the applications of full-spectra solar cells, photodetectors or white light LEDs. But most the 2D alloys showed much lower optoelectronic device performance (such as lower mobility or on/off ration) due to the dopant induced electron scattering.²²⁻²⁴ We have observed that Fe atoms are precisely doped at the sites of Sn atoms, and Fe-SnS₂ showed much higher mobility

and photoresponsivity with respect to pure SnS₂ (Figures 2 and S4, Table S1). This means that the doping method can also provide a choice to explore high mobility and achieve high optoelectronic properties in 2D materials.

Figure 2. Electrical characteristics and photoresponse of Fe_{0.021}Sn_{0.979}S₂ monolayer. (a) Transfer and (b) Output characteristics of Fe_{0.021}Sn_{0.979}S₂. The inset shows the optical image of one typical device and the AFM image of the corresponding sample used for fabricating the device. (c) Time dependent I_{sd} of the transistor based on the Fe_{0.021}Sn_{0.979}S₂ during the light (638 nm, 2.63 μ W) switching on/off under the positive source-drain voltage V_{sd} from 1 to 3 V. (d) Photoresponsivity (R) as function of light power (P) with V_{sd} of 3 V.

Figure S4. Electrical characteristics and photoresponse of SnS₂ monolayer. (a) Transfer and (b) Output characteristics of SnS₂. The inset shows the optical image of one typical device and the AFM image of the corresponding sample used for fabricating the device. (c) Time dependent I_{sd} of the transistor based on the SnS₂ during the light (638 nm, 2.63 μ W) switching on/off under the positive source-drain voltage V_{sd} from 1 to 3 V. (d) Photoresponsivity (R) as function of light power (P) with V_{sd} of 3V.

Table S1. Summary of the field-effect performance transistor parameters including electron mobility (μ), conductivity (σ) of the transistors based on our Fe-SnS₂ and SnS₂ monolayers on Al₂O₃/Si substrate, and the previously reported SnS₂ few-layer on SiO₂/Si substrate.

Parameters	Fe _{0.021} Sn _{0.979} S ₂ monolayer	Fe _{0.015} Sn _{0.985} S ₂ monolayer	Fe _{0.011} Sn _{0.989} S ₂ monolayer	SnS ₂ monolayer	SnS ₂ few-layer ^{3,4}
μ (cm ² /Vs)	8.15	6.24	4.76	3.43	5
ON/OFF ratio	7.3×10^6	6.1×10^6	6.3×10^6	1.3×10^6	4.7×10^6
R (mA/W)	206	\	\	83	8.8

To summarize, we believe that this work is improved obviously with effective evidences on both experimental result and theoretical calculation, and it is very well suited for the audience of “Nature Communications” now. We honestly appreciate your reconsideration about our manuscript.

References.

- 1 Bandurin, D. A. *et al.* High electron mobility, quantum Hall effect and anomalous optical response in atomically thin InSe. *Nat. Nanotechnol.* **12**, 223-227 (2016).
- 2 Wu, J. *et al.* High electron mobility and quantum oscillations in non-encapsulated ultrathin semiconducting Bi₂O₂Se. *Nat. Nanotechnol.* **12**, 530-534 (2017).
- 3 Huang, Y. *et al.* Tin disulfide-an emerging layered metal dichalcogenide semiconductor: materials properties and device characteristics. *ACS Nano* **8**, 10743-10755 (2014).
- 4 Su, G. *et al.* Chemical vapor deposition of thin crystals of layered semiconductor SnS₂ for fast photodetection application. *Nano Lett.* **15**, 506-513 (2015).
- 5 Buscema, M. *et al.* Photocurrent generation with two-dimensional van der Waals semiconductors. *Chem. Soc. Rev.* **44**, 3691-3718 (2015).
- 6 Koppens, F. *et al.* Photodetectors based on graphene, other two-dimensional materials and hybrid systems. *Nat. Nanotechnol.* **9**, 780-793 (2014).
- 7 Gong, C. *et al.* Discovery of intrinsic ferromagnetism in two-dimensional van der Waals crystals. *Nature* **546**, 265-269 (2017).
- 8 Huang, B. *et al.* Layer-dependent ferromagnetism in a van der Waals crystal down to the monolayer limit. *Nature* **546**, 270-273 (2017).
- 9 Sharma, P. *et al.* Ferromagnetism above room temperature in bulk and transparent thin films of Mn-doped ZnO. *Nat. Mater.* **2**, 673-677 (2003).
- 10 Matsumoto, Y. *et al.* Room-temperature ferromagnetism in transparent transition metal-doped titanium dioxide. *Science* **291**, 854-856 (2001).
- 11 Cheng, Y. C., Zhu, Z. Y., Mi, W. B., Guo, Z. B. & Schwingenschlogl, U. Prediction of two-dimensional diluted magnetic semiconductors: Doped monolayer MoS₂ systems. *Phys. Rev. B* **87**, 100401 (2013).
- 12 Mishra, R., Zhou, W., Pennycook, S. J., Pantelides, S. T. & Idrobo, J.-C. Long-range ferromagnetic ordering in manganese-doped two-dimensional dichalcogenides. *Phys. Rev. B* **88**, 144409 (2013).
- 13 Ramasubramaniam, A. & Naveh, D. Mn-doped monolayer MoS₂: An atomically thin dilute magnetic semiconductor. *Phys. Rev. B* **87**, 195201 (2013).
- 14 Yin, L., Mi, W. & Wang, X. Perpendicular magnetic anisotropy and high spin

- polarization in tetragonal Fe₄N/BiFeO₃ heterostructures. *Phys. Rev. Appl.* **6**, 064022 (2016).
- 15 Yin, L., Wang, X. & Mi, W. Perpendicular magnetic anisotropy preserved by orbital oscillation in strained tetragonal Fe₄N/BiFeO₃ bilayers. *ACS Appl. Mater. Interfaces* **9**, 15887-15892 (2017).
- 16 Miyahara, S. & Teranish.T. Magnetic properties of FeS₂ and CoS₂. *J. Appl. Phys.* **39**, 896 (1968).
- 17 Sun, L., Zhou, W., Liang, Y., Liu, L. & Wu, P. Magnetic properties in Fe-doped SnS₂: Density functional calculations. *Comp. Mater. Sci.* **117**, 489-495 (2016).
- 18 Berry, N. *et al.* Atmospheric-pressure chemical vapor deposition of iron pyrite thin films. *Adv. Energy Mater.* **2**, 1124-1135 (2012).
- 19 Liang, S. *et al.* Electrical spin injection and detection in molybdenum disulfide multilayer channel. *Nat. Commun.* **8**, 14947 (2017).
- 20 Xie, L. Two-dimensional transition metal dichalcogenide alloys: preparation, characterization and applications. *Nanoscale* **7**, 18392-18401 (2015).
- 21 Wang, H., Yuan, H., Hong, S. S., Li, Y. & Cui, Y. Physical and chemical tuning of two-dimensional transition metal dichalcogenides. *Chem. Soc. Rev.* **44**, 2664-2680 (2015).
- 22 Zhang, M. *et al.* Two-dimensional molybdenum tungsten diselenide alloys: photoluminescence, raman scattering, and electrical transport. *ACS Nano* **8**, 7130-7137, doi:10.1021/nm5020566 (2014).
- 23 Qingliang, F. *et al.* Growth of large-area 2D MoS_{2(1-x)}Se_{2x} semiconductor alloys. *Adv. Mater.* **26**, 2648-2653, doi:10.1002/adma.201306095 (2014).
- 24 Gong, Y. *et al.* Band gap engineering and layer-by-layer mapping of selenium-doped molybdenum disulfide. *Nano Lett.* **14**, 442-449, doi:10.1021/nl4032296 (2014).

Reviewers' comments:

Reviewer #1 (Remarks to the Author):

The authors have answered my questions. The synthesis of 2d magnetic material from TMDs is encouraging. And they have addressed issues in their transport and photoresponse characterization on the device. I will recommend publication.

Reviewer #2 (Remarks to the Author):

In the revised manuscript, the authors provide more information on the magnetic (and other) properties of their samples, include data from differently doped films and clarify most of the points addressed in the referee reports. In total, I think that the work is improved, however a few points still need attention:

In the reply, the authors explain that they expanded their calculations, taking a larger supercell and applying a Hubbard-U. For SnS2 and FeS2 they give the optimized U-values, but it's unclear what has been chosen for Fe-doped SnS2. In the reply they write that Fig.4 and Fig.S13 were obtained with DFT+U, but in the manuscript and supplement this information is lost. They obtain an energy difference between FM and AFM state of 6.7 meV. According to eq.4 of the reply, this would correspond to $T_C = 51.8$. Taking $K_a = 2.3$ meV from the main text, I get - according to eq.5 in the supplement - $T_C = 34.1$ K. On the other hand, in their reply to comment 4 they use a bulk T_C of 187 K, obviously from ref.[27], take for K_a the estimate of the shape anisotropy (SA) of 2 K ($= 0.17$ meV) and estimate from that $T_C = 34.7$ K. This is similar to the 34.1 K from above, but two errors compensate each other: The bulk T_C is based on a Fe-Fe interaction that is much higher as their distance is closer in [27], the K_a lacks the contribution of the magnetocrystalline anisotropy (KCA) and is, therefore, much smaller. Correct would be $K_a = K_{MCA} + K_{SA} = 2.47$ eV and then eq.5 gives 35.7 K for T_C , in good agreement with experiment.

In the manuscript, line 216, they give $K_a = 2$ K and cite ref.[44] (the source of eq.3) for that, a few lines later they claim that the MAE was obtained by eq.4 and amounts to 2.3 meV, and in the supporting material they write that the MAE was obtained by noncollinear calculations with SOC in second variation. Since not all of these things can be correct, the authors should clean up the text accordingly.

In the supplement, fig.S12 provides interesting information about a 2.7 % Fe doped SnS2 where probably Fe clusters are formed. Although T_C is enhanced, the saturation magnetization is one order of magnitude lower than that of Fe/SnS2 without clusters. Now, this indicates that some antiferromagnetic interaction is probably present, also in [27] it was found that Fe couples AFM across the layers. This is in contrast with the statement in lines 187/188, that interlayer exchange-interaction can be neglected. On the other hand, it indicates that monolayer samples could have much better magnetic properties if the dominant exchange interaction is FM. I think that such issues have to be discussed, also the small saturation magnetization in fig.3 might originate from that. (If available, the authors could show the saturation magnetization of the 1.5% and 1.1% doped samples.)

Another point that needs explanation in Fig.S12 is the note " $H = 1$ T" in panel d and indication of the magnetization direction in panel b (also in Fig.S11).

Finally, I suggest to improve Fig.S1 where the line-scans are barely visible and the scan line should be indicated. Also in Fig.S2d and f it's hard to find a contrast, at least in my printout.

Reviewers' comments:

Reviewer #1 (Remarks to the Author):

The authors have answered my questions. The synthesis of 2d magnetic material from TMDs is encouraging. And they have addressed issues in their transport and photoresponse characterization on the device. I will recommend publication.

Reviewer #2 (Remarks to the Author):

In the revised manuscript, the authors provide more information on the magnetic (and other) properties of their samples, include data from differently doped films and clarify most of the points addressed in the referee reports. In total, I think that the work is improved, however a few points still need attention:

In the reply, the authors explain that they expanded their calculations, taking a larger supercell and applying a Hubbard-U. For SnS2 and FeS2 they give the optimized U-values, but it's unclear what has been chosen for Fe-doped SnS2. In the reply they write that Fig.4 and Fig.S13 were obtained with DFT+U, but in the manuscript and supplement this information is lost. They obtain an energy difference between FM and AFM state of 6.7 meV. According to eq.4 of the reply, this would correspond to $T_C = 51.8$. Taking $K_a = 2.3$ meV from the main text, I get - according to eq.5 in the supplement - $T_C = 34.1$ K. On the other hand, in their reply to comment 4 they use a bulk T_C of 187 K, obviously from ref.[27], take for K_a the estimate of the shape anisotropy (SA) of 2 K ($= 0.17$ meV) and estimate from that $T_C = 34.7$ K. This is similar to the 34.1 K from above, but two errors compensate each other: The bulk T_C is based on a Fe-Fe interaction that is much higher as their distance is closer in [27], the K_a lacks the contribution of the magnetocrystalline anisotropy (KCA) and is, therefore, much smaller. Correct would be $K_a = K_{MCA} + K_{SA} = 2.47$ eV and then eq.5 gives 35.7 K for T_C , in good agreement with experiment.

In the manuscript, line 216, they give $K_a = 2$ K and cite ref.[44] (the source of eq.3) for that, a few lines later they claim that the MAE was obtained by eq.4 and amounts to 2.3 meV, and in the supporting material they write that the MAE was obtained by noncollinear calculations with SOC in second variation. Since not all of these things can be correct, the authors should clean up the text accordingly.

In the supplement, fig.S12 provides interesting information about a 2.7 % Fe doped SnS2 where probably Fe clusters are formed. Although T_C is enhanced, the saturation magnetization is one order of magnitude lower than that of Fe/SnS2 without clusters. Now, this indicates that some antiferromagnetic interaction is probably present, also in [27] it was found that Fe couples AFM across the layers. This is in contrast with the statement in lines 187/188, that interlayer exchange-interaction can be neglected. On the other hand, it indicates that monolayer samples could have much better magnetic properties if the dominant exchange interaction is FM. I think that such issues have to be discussed, also the small saturation magnetization in fig.3 might originate from that. (If available, the authors could show the saturation magnetization of the 1.5% and 1.1% doped samples.)

Another point that needs explanation in Fig.S12 is the note " $H = 1T$ " in panel d and indication of the magnetization direction in panel b (also in Fig.S11).

Finally, I suggest to improve Fig.S1 where the line-scans are barely visible and the scan line should be indicated. Also in Fig.S2d and f it's hard to find a contrast, at least in my printout.

For Reviewer 1:

Dear referee:

We appreciated your insightful suggestions which obviously helped us to improve the quality of our manuscript.

Comments to the author:

The authors have answered my questions. The synthesis of 2D magnetic material from TMDs is encouraging. And they have addressed issues in their transport and photoresponse characterization on the device. I will recommend publication.

Our reply: Thank you for your professional comments and high evaluation of our manuscript. Now, we submit the revised manuscript again, corresponding revisions are made according to all referees and editor's comments.

For Reviewer 2:

Dear referee:

We appreciated your insightful suggestions which helped us to improve the quality of our manuscript. Corresponding revisions were made according to your comments.

Comments to the author:

In the revised manuscript, the authors provide more information on the magnetic (and other) properties of their samples, include data from differently doped films and clarify most of the points addressed in the referee reports. In total, I think that the work is improved, however a few points still need attention:

Our reply: We have carried out groups of new experiments and theoretical calculations according to your comments which obviously help us improve the quality and novelty of the manuscript. The replies and corresponding revisions are as follows:

Comment 1. In the reply, the authors explain that they expanded their calculations, taking a larger supercell and applying a Hubbard-U. For SnS₂ and FeS₂ they give the optimized U-values, but it's unclear what has

been chosen for Fe-doped SnS₂. In the reply they write that Fig.4 and Fig.S13 were obtained with DFT+U, but in the manuscript and supplement this information is lost.

Our reply: Thank you for your kind reminder! The bandgaps of SnS₂ and FeS₂ are 1.61 eV and 0.56 eV calculated by GGA method, respectively. The values are lower than that in experiment. We used GGA+U to calculate the bandgaps and added the information of calculating method in the manuscript.

Our revision:

Wu *et al.* have investigated the magnetism of bulk Fe doped SnS₂ in detail and found that the ferrimagnetism comes from the exchange interaction between doped Fe atoms at the intralayer sites.²⁷ Here, first-principles calculation is used to further investigate the electronic and magnetic properties of Fe-SnS₂ monolayer (**Figure 4a**). By the GGA calculation, the bandgaps of SnS₂ and FeS₂ are 1.61 eV and 0.56 eV, respectively. The bandgaps of SnS₂ and FeS₂ are 2.07 eV and 0.97 eV, which are larger than those in theory. Here, we test a group of simulation with different U values and find that the optimized bandgap of SnS₂ is 2.06 eV (U=8.0) and the bandgap of FeS₂ is 0.95 eV (U=1.8), consistent well with the experimental value.^{43,44}

Comment 2. They obtain a energy difference between FM and AFM state of 6.7 meV. According to eq.4 of the reply, this would correspond to $T_C = 51.8$. Taking $K_a = 2.3$ meV from the main text, I get according to eq.5 in the supplement $T_C = 34.1$ K. On the other hand, in their reply to comment 4 they use a bulk T_C of 187 K, obviously from ref.[27], take for K_a the estimate of the shape anisotropy (SA) of 2 K (= 0.17 meV) and estimate from that $T_C = 34.7$ K. This is similar to the 34.1 K from above, but two errors compensate each other: The bulk T_C is based on a Fe-Fe interaction that is much higher as their distance is closer in [27], the K_a lacks the contribution of the magnetocrystalline anisotropy (KCA) and is, therefore, much smaller. Correct would be $K_a = K_{MCA} + K_{SA} = 2.47$ meV and then eq.5 gives 35.7 K for T_C , in good agreement with experiment.

In the manuscript, line 216, they give $K_a = 2$ K and cite ref.[44] (the source of eq.3) for that, a few lines later they claim that the MAE was obtained by eq.4 and amounts to 2.3 meV, and in the supporting material they write that the MAE was obtained by noncollinear calculations with SOC in second variation. Since not all of these things can be correct, the authors should clean up the text accordingly.

Our reply: Thank you for your professional comment! We have finished some new calculation during this revision, reset the value of K_a

($K_a = K_{mca} + K_{sa} = 2.3 \text{ meV} + 0.17 \text{ meV} = 2.47 \text{ meV}$) and calculated the bulk Curie temperature (T_b) of bulk Fe-SnS₂ ($T_b = 56 \text{ K}$). The Curie temperature T_C of the Fe-SnS₂ monolayer is estimated to be 37 K (eq. 3) in accordance with the experiment result of 31 K. In order to keep the self-consistency of the manuscript, we have removed eq.4 and cleaned up the demonstration of calculating method in supporting material. We have made detailed revisions according to your comments about the calculation of Curie temperature, which can increase the scientific accuracy and clarity of our manuscript. The revisions are shown as follows:

Our revision:

By the GGA calculation, the bandgaps of SnS₂ and FeS₂ are 1.61 eV and 0.56 eV, respectively. The bandgaps of SnS₂ and FeS₂ are 2.07 eV and 0.97 eV, which are larger than those in theory. Here, we test a group of simulation with different U values and find that the optimized bandgap of SnS₂ is 2.06 eV (U=8.0) and the bandgap of FeS₂ is 0.95 eV (U=1.8), consistent well with the experimental value.

The Curie temperature T_C of the Fe-SnS₂ monolayer can be estimated by the relation:⁴⁹

$$T_C = T_b / \ln\left(\frac{3\pi T_b}{4K_a}\right) \quad (3)$$

Where T_b is the bulk Curie temperature, K_a is the anisotropy constant. The bulk magnetic energy ΔE is calculated to be -14.3 meV (**Figure S15**) and T_b is estimated to be 56 K based on the mean-field theory and Heisenberg model.⁵⁰ $K_a = K_{mca} + K_{sa}$, where K_{mca} is the magnetocrystalline anisotropy constant (2.3 meV) and K_{sa} is the shape anisotropy constant (about 0.17 meV^{49}). Thus K_a is 2.47 meV. The calculated T_C is about 37 K, consistent with the experiment result of 31 K.

Figure S15. Atomic structure of bulk Fe-SnS₂. (a) side view (b) top view of the Fe-SnS₂. The calculation is carried out by DFT+U method.

First-principles spin-polarized calculations are performed on the basis of density functional theory (DFT) using projector-augmented wave (PAW) potentials.³ The exchange-correlation interactions are treated by the generalized gradient approximation (GGA) with Perdew–Burke–Ernzerhof (PBE) functional.⁴ The plane-wave cutoff energy is 400 eV. Monkhorst–Pack (MP) meshes of $5 \times 5 \times 1$ and $21 \times 21 \times 1$ are employed for geometry optimization and calculation of density of states, respectively. To obtain reliable values of MAE, dense k points of $21 \times 21 \times 1$ were used

the calculations.

Comment 3. In the supplement, Fig.S12 provides interesting information about a 2.7 % Fe doped SnS₂ where probably Fe clusters are formed. Although T_C is enhanced, the saturation magnetization is one order of magnitude lower than that of Fe/SnS₂ without clusters. Now, this indicates that some antiferromagnetic interaction is probably present, also in [27] it was found that Fe couples AFM across the layers. This is in contrast with the statement in lines 187/188, that interlayer exchange-interaction can be neglected. On the other hand, it indicates that monolayer samples could have much better magnetic properties if the dominant exchange interaction is FM. I think that such issues have to be discussed, also the small saturation magnetization in Fig.3 might originate from that. (If available, the authors could show the saturation magnetization of the 1.5% and 1.1% doped samples.)

Our reply: Thank you for your professional comment! The average magnetic moment of the magnetic atoms is crucial to understand the magnetic mechanism of the sample. Wu et al. have investigated the magnetism of bulk Fe doped SnS₂ and found that the ferrimagnetism comes from the exchange interaction between doped Fe atoms at the intralayer sites and the coupling between doped Fe atoms at the interlayer sites is weak AFM

by calculations.¹ We found that, the statement in previous version (In the layered materials, the exchange interaction between the layers is weak and almost ruled out because of the van der Waals gaps.^{1,2} We can only consider the magnetic property of monolayer Fe-SnS₂.) is insufficient. We have revised the statement according to your suggestions.

We have also carried out groups of new experiments at this revision, and additionally measured the saturation magnetization of the 1.5% and 1.1% doped samples. Based on the experimental results, we found that the Fe_{0.021}Sn_{0.979}S₂ and Fe_{0.015}Sn_{0.985}S₂ samples show ferromagnetic at 2 K, while Fe_{0.011}Sn_{0.989}S₂ sample show paramagnetic at 2 K (**Figure S14**). An average magnetic moment of Fe atom is about 0.020 μ_B per atom for Fe_{0.021}Sn_{0.979}S₂ (**Figure 3b**) and 0.021 μ_B per atom for Fe_{0.015}Sn_{0.985}S₂ (**Figure S14**) at 2 K. The magnetic moment of Fe atom is low, and this phenomenon should result from the AFM coupling. Based on the reported theoretical results, the AFM coupling should most come from the doped Fe atoms at the intralayer and interlayer sites.^{1,3-5}

Our revision:

Fe_{0.021}Sn_{0.979}S₂ and Fe_{0.015}Sn_{0.985}S₂ show ferromagnetic at 2 K, while Fe_{0.011}Sn_{0.989}S₂ show paramagnetic at 2 K (**Figure S14**). The result demonstrates that magnetic property of Fe-SnS₂ varies with the Fe concentration.

Wu *et al.* have investigated the magnetism of bulk Fe doped SnS₂ in detail and found that the ferrimagnetism comes from the exchange interaction between doped Fe atoms at the intralayer sites.²⁷ Here, first-principles calculation is used to further investigate the electronic and magnetic properties of Fe-SnS₂ monolayer (**Figure 4a**). By the GGA calculation, the bandgaps of SnS₂ and FeS₂ are 1.61 eV and 0.56 eV, respectively. The bandgaps of SnS₂ and FeS₂ are 2.07 eV and 0.97 eV, which are larger than those in theory. Here, we test a group of simulation with different U values and find that the optimized bandgap of SnS₂ is 2.06 eV (U=8.0) and the bandgap of FeS₂ is 0.95 eV (U=1.8), consistent well with the experimental value.^{43,44}

An average magnetic moment of Fe atom is about 0.020 μ_B per atom for Fe_{0.021}Sn_{0.979}S₂ (**Figure 3b**) and 0.021 μ_B per atom for Fe_{0.015}Sn_{0.985}S₂ (**Figure S14**) at 2 K. The magnetic moment of Fe atom is relatively low,⁵¹ and this phenomenon should result from the AFM coupling between doped Fe atoms at the intralayer and interlayer sites. It is reported that in the magnetic atom doped 2D TMDs, the difference in energies between FM and AFM coupling is related with the distances of magnetic atoms. When the distance between two magnetic atoms is long enough, it will be AFM coupling.^{25,26,29} Wu *et al.* have demonstrated that in the Fe-SnS₂, Fe

atoms can be AFM coupling across the layers. In the Fe-SnS₂ crystal,²⁷ Fe atoms are distributed randomly (Figure 1g), the AFM coupling should be existed and decrease the average magnetic moment of Fe atoms.

Figure S14. (a) and (b) Magnetic hysteresis loops for Fe_{0.015}Sn_{0.985}S₂ and Fe_{0.011}Sn_{0.989}S₂ at 2 K, respectively. The applied magnetic field is perpendicular to the sheet.

Comment 4. Another point that needs explanation in Fig.S12 is the note "H = 1T" in panel d and indication of the magnetization direction in panel b (also in Fig.S11).

Our reply: Thank you for your kind reminder! "H = 1T" in Fig. S12 means the applied magnetic field is 1T. The applied magnetic field is perpendicular to the sheet (H_{\perp}) in Fig.S11 and Fig.S12.

Our revision:

Figure S11. Magnetic hysteresis loops for $\text{Fe}_{0.021}\text{Sn}_{0.979}\text{S}_2$ bulk at 40 K, 35 K, 30 K and 25 K using VSM, respectively. The applied magnetic field is perpendicular to the sheet (H_{\perp}).

Figure S12. (a) XPS spectra of Fe-SnS_2 with Fe cluster. (b) Magnetic

hysteresis loops Fe-SnS₂ with Fe cluster at 300 K using VSM. (c) The expanded view of the loop in (b). (d) Magnetization as a function of temperature for the Fe-SnS₂ with Fe cluster from 300 to 450 K. The applied magnetic field is 1T. The applied magnetic field is perpendicular to the sheet (H_{\perp}).

Comment 5. Finally, I suggest to improve Fig.S1 where the line-scans are barely visible and the scan line should be indicated. Also in Fig.S2d and f it's hard to find a contrast, at least in my printout.

Our reply: Thank you for your kind reminder! We have added the line-scans in Fig.S1 and improved the contrast in Fig.S2d.

Our revision:

Figure S1. (a) MFM image of Fe_{0.021}Sn_{0.979}S₂ flake. (b) AFM and (c) MFM images of pure SnS₂ flake. The height and phase shift are obtained along the white dotted line.

Figure S2. TEM images and EDS elemental mapping images of $\text{Fe}_{0.021}\text{Sn}_{0.979}\text{S}_2$ flake. (a) Low-magnification TEM image, (b) SAED pattern and (c) High-magnification TEM image of $\text{Fe}_{0.021}\text{Sn}_{0.979}\text{S}_2$ flake. (d)-(f) EDS elemental mapping images of Sn, S, Fe from the marked area in (a), respectively.

Reference:

- 1 Sun, L., Zhou, W., Liang, Y., Liu, L. & Wu, P. Magnetic properties in Fe-doped SnS_2 : Density functional calculations. *Comp. Mater. Sci.* **117**, 489-495 (2016).
- 2 Buhannic, M. A., Danot, M., Colombet, P., Dordor, P. & Fillion, G. Thermopower and low-dc-field magnetization study of the layered Fe_xZrSe_2 compounds: Anderson-type localization and anisotropic spin-glass behavior. *Phys. Rev. B* **34**, 4790-4795 (1986).
- 3 Cheng, Y. C., Zhu, Z. Y., Mi, W. B., Guo, Z. B. & Schwingenschlogl, U. Prediction of two-dimensional diluted magnetic semiconductors: Doped monolayer MoS_2 systems. *Phys. Rev. B* **87**, 100401 (2013).
- 4 Mishra, R., Zhou, W., Pennycook, S. J., Pantelides, S. T. & Idrobo, J.-C. Long-range ferromagnetic ordering in manganese-doped two-dimensional dichalcogenides. *Phys. Rev. B* **88**, 144409 (2013).
- 5 Ramasubramaniam, A. & Naveh, D. Mn-doped monolayer MoS_2 : An atomically thin dilute magnetic semiconductor. *Phys. Rev. B* **87**, 195201 (2013).

REVIEWERS' COMMENTS:

Reviewer #1 (Remarks to the Author):

The revised version of the manuscript looks good. I will recommend publication.

Reviewer #2 (Remarks to the Author):

The authors revised the manuscript appropriately, the only point to check is the shape anisotropy: They cite ref.[49] where there is a rough estimate that $K_{SA} = 2 \cdot 10^{-4}$ G but - as usual for thin films - favoring an in-plane magnetization. Therefore, Bander and Mills subtract that value from the spin-orbit contribution. If the sign of K_{SA} is really positive, it's OK as it is, otherwise it should be corrected. When they give the value of U for the DFT+U calculation, it's also useful to specify the atom and shell where the U acts on.

For Reviewer 2:

Dear referee:

We appreciated your insightful suggestions which helped us to improve the quality of our manuscript. Corresponding revisions were made according to your comments.

Comments to the author:

The authors revised the manuscript appropriately, the only point to check is the shape anisotropy: They cite ref.[49] where there is a rough estimate that $K_{SA} = 2 \cdot 10^{-4}$ G but - as usual for thin films - favoring an in-plane magnetization. Therefore, Bander and Mills subtract that value from the spin-orbit contribution. If the sign of K_{SA} is really positive, it's OK as it is, otherwise it should be corrected. When they give the value of U for the DFT+U calculation, it's also useful to specify the atom and shell where the U acts on.

Our reply: Thank you for your professional comments. Magnetic anisotropy in thin films has attracted a lot of attention in the past few decades. The easy axis results from the competition between shape

anisotropy and magnetocrystalline anisotropy. $K_a = K_{mca} + K_{sa}$, where K_a is the anisotropy constant, K_{mca} is the magnetocrystalline anisotropy constant and K_{sa} is the shape anisotropy constant. When K_a is positive, the easy axis is normal to the film, otherwise, the easy axis is parallel to the film. It is reported that Fe and Co atoms become ultrathin film, their easy axis will be normal to the film.^{1,2} In the Ref. [49],³ Bander and Mills investigated two-dimensional Heisenberg ferromagnet and its easy axis is normal to the film. As the system we studied is similar with theirs, and they subtracted K_{sa} from the spin-orbit contribution,³ we have revised the K_{sa} as -0.17 meV, and the K_a is 2.13 meV, the Curie temperature is estimated about 33 K.

We have used DFT+U to obtain the band structure and magnetic property of Fe doped SnS₂. The U acted on the d orbitals of Fe (U=) and Sn (U=) atoms.

Our revision: The Curie temperature, T_C , of the Fe-SnS₂ monolayer can be estimated by the relation:⁴⁹

$$T_C = T_b / \ln\left(\frac{3\pi T_b}{4K_a}\right) \quad (3)$$

where T_b is the bulk Curie temperature, and K_a is the anisotropy constant. The bulk magnetic energy, ΔE , was calculated to be -14.3 meV (**Supplementary Fig. 15**), and the T_b was estimated to be 56 K based on the mean-field theory and Heisenberg model.⁵⁰ $K_a = K_{mca} + K_{sa}$, where K_{mca}

is the magnetocrystalline anisotropy constant (2.3 meV), and K_{sa} is the shape anisotropy constant (approximately -0.17 meV⁴⁹). Thus, K_a is 2.13 meV. The calculated T_C is approximately 33 K, which is consistent with the experimental result of 31 K.

DFT calculations. According to the GGA calculations, the bandgaps of SnS₂ and FeS₂ are 1.61 eV and 0.56 eV, respectively. The bandgaps of SnS₂ and FeS₂ are 2.07 eV and 0.97 eV, which are larger than those determined via the calculations. We tested different U values and found that the bandgap of SnS₂ is 2.06 eV (U=8.0 on the d orbital) and the bandgap of FeS₂ is 0.95 eV (U=1.8 on the d orbital), which are consistent with the experimental values. First-principles spin-polarized calculations were performed on the basis of density functional theory (DFT) using project or augmented wave (PAW) potentials.⁵² The exchange-correlation interactions were treated by the generalized gradient approximation (GGA) with the Perdew–Burke–Ernzerhof (PBE) functional.⁵³ The plane-wave cutoff energy was 400 eV. Monkhorst–Pack (MP) meshes of 5×5×1 and 21×21×1 were employed for geometry optimization and the calculation of density of states, respectively. To obtain reliable values for the MAEs, dense k points of 21×21×1 were used for the calculations.

Reference

- 1 Jonker, B. T., Walker, K. H., Kisker, E., Prinz, G. A. & Carbone, C.

Spin-polarized photoemission study of epitaxial Fe(001) films on Ag(001). *Phys. Rev. Lett.* **57**, 142-145 (1986).

- 2 Chappert, C., Dang, K. L., Beauvillain, P., Hurdequint, H. & Renard, D. Ferromagnetic resonance studies of very thin cobalt films on a gold substrate. *Phys. Rev. B* **34**, 3192-3197 (1986).
- 3 Bander, M. & Mills, D. L. Ferromagnetism of ultrathin films. *Phys. Rev. B* **38**, 12015-12018 (1988).